# Probabilistic Missing Value Imputation
# for Mixed Categorical and Ordered Data

**Yuxuan Zhao**
Cornell University
yz2295@cornell.edu

**Alex Townsend**
Cornell University
townsend@cornell.edu

**Madeleine Udell**
Stanford University
udell@stanford.edu

## Abstract

Many real-world datasets contain missing entries and mixed data types including categorical and ordered (e.g. continuous and ordinal) variables. Imputing the missing entries is necessary, since many data analysis pipelines require complete data, but this is challenging especially for mixed data. This paper proposes a probabilistic imputation method using an *extended Gaussian copula model* that supports both single and multiple imputation. The method models mixed categorical and ordered data using a latent Gaussian distribution. The unordered characteristics of categorical variables is explicitly modeled using the argmax operator. The method makes no assumptions on the data marginals nor does it require tuning any hyperparameters. Experimental results on synthetic and real datasets show that imputation with the extended Gaussian copula outperforms the current state-of-the-art for both categorical and ordered variables in mixed data.

## 1  Introduction

Modern datasets from healthcare, the sciences, and the social sciences often contain missing entries and mixed data types such as continuous, ordinal, and categorical. Social survey datasets, for example, are typically mixed because they include variables like age (continuous), demographic group (categorical), and Likert scales (ordinal) measuring how strongly a respondent agrees with certain stated opinions. Continuous variables are encoded as real numbers and sometimes called numeric. We refer to variables that admit a total order (e.g. continuous and ordinal) as *ordered* variables. In contrast, a categorical variable, also called nominal, can take one of a fixed number of *unordered values* such as "A", "B", "AB", or "O" for blood type.

Most data analysis techniques require a complete dataset, so missing data imputation is an essential preprocessing step. It is also often of interest to propagate imputation uncertainty into subsequent analyses through multiple imputation, which generates several potentially different imputed datasets [25]. An imputation method should ideally use all the collected data—regardless of data type—to impute any missing data entries. However, most imputation approaches, whether explicitly or implicitly, assume that each variable admits a total order and, as a result, cannot impute categorical variables without proper preprocessing.

There is no satisfying, successful, and widely adopted method for imputing categorical variables, especially in mixed datasets. It is tempting to reduce categorical imputation to ordinal imputation using an *integer encoding* in which each category is assigned a number; however, this encoding requires choosing an arbitrary ordering for the categories that may affect the results of downstream (e.g., predictive) models. (This problem is most severe for linear and neural network models, whereas tree-based models are less sensitive [31].) Instead, it is more common to use *one-hot encoding* to represent a categorical variable using a binary vector with one entry for each category. With this encoding, imputation methods developed for continuous or binary data, such as the popular low rank matrix completion methods (LRMC) [24, 17, 29], can be jerry-rigged for categorical imputation.

36th Conference on Neural Information Processing Systems (NeurIPS 2022).

However, LRMC is only proper when the encoded matrix (both ordered and encoded categorical variables) is approximately low rank. It has been observed that LRMC performs poorly on long skinny (many samples $n$ and few features $p$) datasets because the low rank assumption fails [9, 35].

Iterative imputation methods including MICE [4] and missForest [27] can directly operate on the unordered categories. MICE learns the conditional distribution of each variable (whether ordered or unordered) using all other variables via linear and logistic regression. However, the learned conditional models may be incompatible in the sense that the joint distribution cannot exist [4]. MissForest uses random forest to predict each variable and yields more accurate single imputations [27], but cannot provide multiple imputation. Both methods converge slowly (or sometimes diverge) for large datasets because they train many models on (possibly) ill-conditioned data.

This paper expands on the idea of modeling categorical data and multivariate data interaction using a latent continuous space [11, 6]. This approach can explicitly model the categorical distribution, which is critical for providing multiple imputation. This paper models the latent space as a Gaussian distribution and models each categorical variable as the argmax of a latent Gaussian vector. This choice is inspired by the Gaussian copula model [14, 8, 10], which yields high quality imputations for ordered data [35]. See [36] for a concise review and [32] for comprehensive methodology about Gaussian copula imputation. However, it cannot be used when the data contains categorical variables. This paper proposes the *extended Gaussian copula* to overcome this limitation by explicitly modeling each data type. In contrast, other imputation methods typically require bespoke preprocessing for the input data to perform well. For mixed categorical and ordered data, the extended Gaussian copula model generates each categorical variable using a latent Gaussian vector and each ordered variable using a latent Gaussian scalar. The latent Gaussian correlations capture the dependence structure.

**Contributions.**    This paper makes three main contributions to the literature:

1. A probabilistic model, the *extended Gaussian copula*, for mixed data including continuous, ordinal, and categorical variables. The model is free of hyperparameters and makes no assumptions on the marginal distribution of each data type.

2. A *single imputation* method that empirically provides state-of-the-art accuracy for mixed data and a *multiple imputation* method that can quantify imputation uncertainty by measures such as the category probability for categorical variables and confidence intervals for continuous variables.

3. A robust and efficient parameter fitting algorithm for the extended Gaussian copula that supports further acceleration through parallelization, mini-batch training, and low rank structure.

**Related work.**    Modeling a categorical variable as the argmax of a latent continuous vector is a classical technique used in the multinomial probit model [3, 22] in the context of supervised learning. Instead, our model is more similar to [6], which proposed one way to accommodate categorical variables in the Gaussian copula. However, the model in [6] has many redundant parameters and thus is unidentifiable, while the extended Gaussian copula is identifiable and can match any categorical marginal distribution.

## 2    Methodology

**Notation.**    For $f : \mathbb{R}^K \to \mathbb{R}$, we define the (possibly set-valued) pre-image of $f$ as $f^{-1}(y) := \{x : f(x) = y\}$. We define $\mathrm{argmax}(\mathbf{z})$ of a vector $\mathbf{z} = (z_1, \ldots, z_p)$ as the index of the maximum entry so that $\mathbf{z}_{\mathrm{argmax}(\mathbf{z})} = \max_{i=1,\ldots,p} z_i$. We use $\mathcal{N}(\boldsymbol{\mu}, \Sigma)$ to denote the multivariate normal distribution with mean $\boldsymbol{\mu}$ and covariance $\Sigma$. We use $\mathbf{0}$ to denote the all-zero vector and $\mathbf{1}$ for the all-one vector, where the context determines the length of the vector.

We introduce our model in Section 2.1, then derive the model-based imputation methods in Section 2.2 and finally present model estimation algorithms in Section 2.3. Throughout the paper, we assume the missing complete at random (MCAR) mechanism: missing values are uniform and independent of any data. Nevertheless, we show our method performs reasonably well under missing at random (MAR) and missing not at random (MNAR) assumptions through experiments. We also discuss how violating the MCAR assumption may affect the assumptions of the proposed model in Section 4.

## 2.1 Extended Gaussian copula with categorical variables

We first show how to model a categorical variable by transforming a latent Gaussian vector. Then we extend this model to generate categorical vectors and mixed categorical and ordered vectors. To ease the notation, we assume that all categorical variables have $K$ categories encoded as $\{\text{"1"}, \ldots, \text{"K"}\}$. It is straightforward to allow categorical variables with different numbers of categories.

### 2.1.1 Univariate categorical variable

We model a univariate categorical variable $x$ with $K$ categories as the argmax of a $K$-dim latent Gaussian $\mathbf{z} = (z_1, \ldots, z_K)$ with some mean $\boldsymbol{\mu} = (\mu_1, \ldots, \mu_K)$ and identity covariance. That is,

$$x := \operatorname{argmax}(\mathbf{z} + \boldsymbol{\mu}), \qquad \mathbf{z} \sim \mathcal{N}(\mathbf{0}, \mathbf{I}_K), \tag{1}$$

We call the distribution in Eq. (1) the *Gaussian-Max* distribution. Without loss of generality, we assume $\mu_1 = 0$, as the argmax is invariant under translations, i.e., $\mu \leftarrow \mu + \alpha \mathbf{1}$ for $\alpha \in \mathbb{R}$. While a dense covariance matrix is sometimes used for $\mathbf{z}$ [6], we prefer the model of Eq. (1) as it is identifiable. Theorem 1 states that any categorical distribution corresponds to a unique choice of $\mu$ under Eq. (1). All proofs are in the supplement. In Section 2.3.1 we describe an algorithm to estimate $\boldsymbol{\mu}$ for a given categorical distribution.

**Theorem 1** (Existence and Uniqueness). *For any categorical distribution $\mathbb{P}[x = \text{"k"}] = p_k > 0$ for $k = 1, \ldots, K$ such that $\sum_{k=1}^{K} p_k = 1$, there is a unique $\boldsymbol{\mu} \in \mathbb{R}^K$ with $\mu_1 = 0$ such that*

$$\mathbb{P}_{\mathbf{z} \sim \mathcal{N}(\mathbf{0}, \mathbf{I}_K)}[\operatorname{argmax}(\mathbf{z} + \boldsymbol{\mu}) = k] = p_k, \quad k = 1, \ldots, K. \tag{2}$$

### 2.1.2 Multivariate categorical vector

We model a categorical vector $\mathbf{x} = (x_1, \ldots, x_p)$ by supposing each of its entries $x_j$ follows the Gaussian-Max distribution. That is, $x_j = \operatorname{argmax}(\mathbf{z}^{(j)} + \boldsymbol{\mu}^{(j)})$ for some $\boldsymbol{\mu}^{(j)}$ and isotropic Gaussian $\mathbf{z}^{(j)}$. Additionally, the latent Gaussian variables $\mathbf{z}^{(j)}$ corresponding to different categorical variables may be correlated. We call this model the *categorical latent Gaussian (CLG)* model (see Definition 1). For $\mathbf{z} := (\mathbf{z}^{(1)}, \ldots, \mathbf{z}^{(p)})$, we use $[j]$ to denote the indices of $\mathbf{z}^{(j)}$ in $\mathbf{z}$, so $\mathbf{z}_{[j]} = \mathbf{z}^{(j)}$ for $j = 1, \ldots, p$.

**Definition 1** (Categorical latent Gaussian). *For a categorical vector $\mathbf{x} = (x_1, \ldots, x_p)$, we say $\mathbf{x}$ follows the categorical latent Gaussian $\mathbf{x} \sim \operatorname{CLG}(\Sigma, \boldsymbol{\mu})$, if there exists a correlation matrix $\Sigma$ and $\boldsymbol{\mu}$ such that (1) for $\mathbf{z} \sim \mathcal{N}(\mathbf{0}, \Sigma)$, $x_j = \operatorname{argmax}(\mathbf{z}_{[j]} + \boldsymbol{\mu}_{[j]})$; (2) $\Sigma_{[j],[j]} = \mathbf{I}_K$, for every $j = 1, \ldots, p$.*

In the CLG model, the value of $\boldsymbol{\mu}_{[j]}$ suffices to determine the marginal distribution of each categorical $x_j$, as the marginal distribution of $\mathbf{z}_{[j]}$ is $\mathcal{N}(\mathbf{0}, \mathbf{I}_K)$ and independent of $\Sigma$. The correlation matrix $\Sigma$ introduces dependencies between different categorical variables in $\mathbf{x}$.

Consider two categorical variables $x_{j_1}$ and $x_{j_2}$ whose joint distribution is described by $\mathbf{P} \in \mathbb{R}^{K \times K}$, where $p_{kl} = \mathbb{P}[x_{j_1} = k, x_{j_2} = l]$. The CLG model captures the dependence between $x_{j_1}$ and $x_{j_2}$ by the correlation submatrix $\Sigma_{[j_1],[j_2]}$. Note that $\mathbf{P}$ has only $(K-1)^2$ free parameters as the rows and columns must sum to the associated marginal probabilities. This means that $\Sigma_{[j_1],[j_2]}$ in the CLG model is not identifiable. We develop an identifiable variant of the CLG model in the supplement that uses only $(K-1)^2$ parameters in $\Sigma_{[j_1],[j_2]}$. Both models share similar imputation performance, but the identifiable model is not invariant under permutation of the categorical labels while the model in Definition 1 is invariant. For the rest of this paper, we use the permutation-invariant CLG model.

### 2.1.3 Mixed categorical and ordered vector

We model mixed data that contains both categorical and ordered variables by combining the CLG model for categorical variables with the Gaussian copula model for ordered variables [10, 35].

To model $\mathbf{x}$ with ordered variables, the Gaussian copula assumes $\mathbf{x} \sim \operatorname{GC}(\Sigma, \mathbf{f})$ is generated as an elementwise transformed Gaussian, i.e., $\mathbf{x} = \mathbf{f}(\mathbf{z}) = (f_1(z_1), \ldots, f_p(z_p))$, where each $f_j$ is a monotonic increasing function and $\mathbf{z} \sim \mathcal{N}(\mathbf{0}, \Sigma)$. Denoting the cumulative distribution function (CDF) of $x_j$ as $F_j$, the transformation $f_j$ is uniquely determined as $f_j = F_j^{-1} \circ \Phi$, where $\Phi$ denotes the standard Gaussian CDF and $F_j^{-1}(y) := \inf\{x \in \mathbb{R} : F_j(x) \geq y\}$. Specifically, ordinals result from thresholding the latent Gaussian variable. For an ordinal $x_j$, the transformation $f_j(z)$ has the

form $\sum_{s \in S} \mathbb{1}(z > s)$ where $\mathbb{1}(z > s)$ is 1 if $z > s$ and 0 otherwise. Here, the set $S$ is determined by $F_j$. The CLG model has the same form as $\mathbf{x} = \mathbf{f}(\mathbf{z})$ by writing

$$x_j = f_j(\mathbf{z}_{[j]}; \boldsymbol{\mu}_{[j]}) := \mathrm{argmax}(\mathbf{z}_{[j]} + \boldsymbol{\mu}_{[j]}). \tag{3}$$

Hence, we propose the *extended Gaussian copula (EGC)* model.

**Definition 2** (Extended Gaussian copula). *Write a mixed data vector as* $\mathbf{x} = (\mathbf{x}_{\mathrm{cat}}, \mathbf{x}_{\mathrm{ord}})$ *where* $\mathbf{x}_{\mathrm{cat}}$ *collects all categorical variables and* $\mathbf{x}_{\mathrm{ord}}$ *collects all ordered variables. We say* $\mathbf{x}$ *follows the extended Gaussian copula* $\mathbf{x} \sim \mathrm{EGC}(\Sigma, \mathbf{f}^{\mathrm{ord}}, \boldsymbol{\mu})$ *if there exists a correlation matrix* $\Sigma$*, an elementwise monotone* $\mathbf{f}^{\mathrm{ord}}$*, and a* $\boldsymbol{\mu}$ *such that* $\mathbf{x}_{\mathrm{cat}} \sim \mathrm{CLG}(\Sigma_{\mathrm{cat, cat}}, \boldsymbol{\mu})$ *and* $\mathbf{x}_{\mathrm{ord}} \sim \mathrm{GC}(\Sigma_{\mathrm{ord,ord}}, \mathbf{f}^{\mathrm{ord}})$*, where* $\Sigma = \begin{bmatrix} \Sigma_{\mathrm{cat, cat}} & \Sigma_{\mathrm{cat, ord}} \\ \Sigma_{\mathrm{ord, cat}} & \Sigma_{\mathrm{ord,ord}} \end{bmatrix}$.

We can also write $\mathbf{x} = \mathbf{f}(\mathbf{z}) = (\mathbf{f}^{\mathrm{cat}}(\mathbf{z}_{\mathrm{cat}}; \boldsymbol{\mu}), \mathbf{f}^{\mathrm{ord}}(\mathbf{z}_{\mathrm{ord}}))$ for $\mathbf{z} = (\mathbf{z}_{\mathrm{cat}}, \mathbf{z}_{\mathrm{ord}}) \sim \mathcal{N}(\mathbf{0}, \Sigma)$ where $\mathbf{f}$ is defined by all transformation parameters, i.e., $\mathbf{f}^{\mathrm{ord}}$ and $\boldsymbol{\mu}$. Unlike the original Gaussian copula, the dimension of the latent $\mathbf{z}$ does not match that of the data $\mathbf{x}$; instead, it is larger as subvectors of $\mathbf{z}$ correspond to categorical entries of $\mathbf{x}$. The correlation matrix $\Sigma$ must also obey constraints on the submatrix $\Sigma_{\mathrm{cat, cat}}$ according to Definition 1.

## 2.2 Missing data imputation

Now we show how to impute missing values given an EGC model with known parameters. For model parameter estimation, see Section 2.3.

Note each categorical $x_j$ corresponds to $K$ consecutive entries in $\mathbf{z}$: the subvector $\mathbf{z}_{[j]}$ that generates $x_j$. We use $[I]$ to denote the set of indices in $\mathbf{z}$ that corresponds to a set of indices $I$ from $\mathbf{x}$. For example, $[j]$ follows this notation with $I = \{j\}$. Two sets of indices are particularly important: the observed entries $\mathcal{O}$ and missing entries $\mathcal{M}$ for a given mixed data vector $\mathbf{x}$. Thus $[\mathcal{O}]$ and $[\mathcal{M}]$ are the latent dimensions in $\mathbf{z}$ that generate $\mathbf{x}_{\mathcal{O}}$ and $\mathbf{x}_{\mathcal{M}}$, respectively.

Our imputation strategy follows [35]. First, the algorithm identifies a region of the latent space $\mathbf{z}_{[\mathcal{O}]}$ constrained by the observed entries. Then, it imputes the latent dimensions by exploiting the Gaussian distribution of $\mathbf{z}_{[\mathcal{M}]}$ conditional on $\mathbf{z}_{[\mathcal{O}]}$. Finally, the algorithm uses the marginal transformation $\mathbf{f}$ to produce imputations in the data space $\mathbf{x}_{\mathcal{M}}$. Pictorially, we can visualize this process as

$$\mathbf{x}_{\mathcal{O}} \xrightarrow{\mathbf{f}_{\mathcal{O}}^{-1}} \mathbf{z}_{[\mathcal{O}]} \xrightarrow{\Sigma} \mathbf{z}_{[\mathcal{M}]} \xrightarrow{\mathbf{f}_{\mathcal{M}}} \mathbf{x}_{\mathcal{M}}.$$

**Multiple imputation.** Multiple imputation creates several imputed datasets by sampling from the distribution of missing entries conditional on the observations. For the EGC model, Algorithm 1 samples from the distribution of $\mathbf{x}_{\mathcal{M}}$ using two facts: (1) given $\mathbf{x}_{\mathcal{O}}$, the variable $\mathbf{z}_{[\mathcal{O}]}$ follows a truncated Gaussian distribution truncated to $\mathbf{f}_{\mathcal{O}}^{-1}(\mathbf{x}_{\mathcal{O}}) = \prod_{j \in \mathcal{O}} f_j^{-1}(x_j)$; and (2) the random variable $\mathbf{z}_{[\mathcal{M}]} | \mathbf{z}_{[\mathcal{O}]}$ is Gaussian distributed. Section 2.3.3 shows how to sample from the truncated Gaussian random variable given by $\mathbf{z}_{[\mathcal{O}]} | \mathbf{x}_{\mathcal{O}}$. Using the empirical distribution of drawn samples, we can estimate the category probability for a missing categorical variable and build confidence intervals for a missing continuous variable.

---

**Algorithm 1** Multiple imputation via the extended Gaussian Copula

---

1: **Input:** # of imputations $m$, data vector $\mathbf{x}$ observed at $\mathcal{O}$, model parameters $\mathbf{f}$ and $\Sigma$.
2: **for** $s = 1, 2, \ldots, m$ **do**
3:      Sample $\hat{\mathbf{z}}_{[\mathcal{O}]}^{(s)} \sim \mathbf{z}_{[\mathcal{O}]} | \mathbf{x}_{\mathcal{O}}$ :    $\mathcal{N}(\mathbf{0}, \hat{\Sigma}_{[\mathcal{O}],[\mathcal{O}]})$ truncated to $\mathbf{f}_{\mathcal{O}}^{-1}(\mathbf{x}_{\mathcal{O}})$
4:      Sample $\hat{\mathbf{z}}_{[\mathcal{M}]}^{(s)} \sim \mathbf{z}_{[\mathcal{M}]} | \mathbf{z}_{[\mathcal{O}]}$ :    $\mathcal{N}\left( \Sigma_{[\mathcal{M}],[\mathcal{O}]} \hat{\mathbf{z}}_{[\mathcal{O}]}^{(s)}, \Sigma_{[\mathcal{M}],[\mathcal{M}]} - \Sigma_{[\mathcal{M}],[\mathcal{O}]} \Sigma_{[\mathcal{O}],[\mathcal{O}]}^{-1} \Sigma_{[\mathcal{O}],[\mathcal{M}]} \right)$
5:      Compute $\hat{\mathbf{x}}_{\mathcal{M}}^{(s)} = \mathbf{f}_{\mathcal{M}}(\hat{\mathbf{z}}_{[\mathcal{M}]}^{(s)})$
6: **end for**
7: **Output:** $\{\hat{\mathbf{x}}_{\mathcal{M}}^{(s)} | s = 1, .., m\}$.

---

**Single imputation.** To provide an accurate single imputation under the EGC model, we impute based on the conditional mean of the latent Gaussian, i.e.,

$$\hat{\mathbf{x}}_{\mathcal{M}} = \mathbf{f}_{\mathcal{M}}(\mathbb{E}[\mathbf{z}_{[\mathcal{M}]}|\mathbf{x}_{\mathcal{O}}, \Sigma, \mathbf{f}]) = \mathbf{f}_{\mathcal{M}}(\Sigma_{[\mathcal{M}],[\mathcal{O}]}\Sigma_{[\mathcal{O}],[\mathcal{O}]}^{-1}\mathbb{E}[\mathbf{z}_{[\mathcal{O}]}|\mathbf{x}_{\mathcal{O}}, \Sigma, \mathbf{f}]). \tag{4}$$

To compute $\mathbb{E}[\mathbf{z}_{[\mathcal{O}]}|\mathbf{x}_{\mathcal{O}}, \Sigma, \mathbf{f}]$, see Section 2.3.3. With a known mean $\mathbb{E}[\mathbf{z}_{[\mathcal{O}]}|\mathbf{x}_{\mathcal{O}}, \Sigma, \mathbf{f}]$ and marginal distribution $\mathbf{f}_{\mathcal{M}}$, computing the imputation $\hat{\mathbf{x}}_{\mathcal{M}}$ is straightforward. Specifically, for categorical missing $x_j$, we first compute the $K$-dim $\hat{\mathbf{z}}_{[j]} := \mathbb{E}[\mathbf{z}_{[j]}|\mathbf{x}_{\mathcal{O}}, \Sigma, \mathbf{f}]$ and then impute as $\hat{x}_j = \mathrm{argmax}(\hat{\mathbf{z}}_{[j]} + \boldsymbol{\mu}_{[j]})$.

**Online imputation.** Online imputation is the task to impute missing entries in data streams arriving at different times. We can conduct online imputation using the EGC model following [33]: upon observing an incomplete data batch, impute the missing entries with a current saved model, and then update the model using a new observation. See Appendix A on how to update the model.

## 2.3 Parameter estimation

Parameter estimation for the EGC model consists of two steps: (1) we form an estimate $\hat{\mathbf{f}}$ of the marginal transformation $\mathbf{f}$, and (2) we estimate the copula correlation $\Sigma$ given $\hat{\mathbf{f}}$. The marginal transformation differs for ordered and categorical variables. For an ordered $x_j$, since $f_j = F_j^{-1} \circ \Phi$, we can simply estimate $F_j^{-1}$ using the empirical quantile function on the observed entries of $x_j$ as in [20, 35]. For a categorical variable $x_j$, we show how to estimate $f_j(\cdot; \boldsymbol{\mu}_{[j]})$ in Section 2.3.1 and how to estimate $\Sigma$ in Eq. (3) by extending estimation algorithms from [35].

### 2.3.1 Marginal estimation for categorical variables

Here we estimate the mean $\boldsymbol{\mu}$ for which the EGC model, $\mathrm{EGC}(\Sigma, \mathbf{f}^{\mathrm{ord}}, \boldsymbol{\mu})$, matches the observed categorical marginal distribution. Each subvector $\boldsymbol{\mu}_{[j]}$ corresponding to categorical $x_j$ may be estimated independently of all the others, so we drop the variable subscript $j$ to ease the notation.

We must find $\boldsymbol{\mu}$ in Eq. (2) so that the resulting category frequency $p_k$ matches the observed frequency, subject to $\mu_1 = 0$: this setup gives $K - 1$ nonlinear equations with $K - 1$ unknowns. Unfortunately, there is no closed form solution. Thus we resort to iterative root finding algorithms, which use as oracles the value and derivative of $\mathbb{P}[\mathrm{argmax}(\mathbf{z} + \boldsymbol{\mu}) = k]$ in Eq. (2). We use two approximation techniques to evaluate these oracles efficiently.

First we approximate the probability of argmax as the expectation of softmax by noticing

$$\mathbb{P}_{\mathbf{z} \sim \mathcal{N}(\mathbf{0}, \mathbf{I}_K)}[\mathrm{argmax}(\mathbf{z} + \boldsymbol{\mu}) = k] = \lim_{\beta \to \infty} \mathbb{E}_{\mathbf{z} \sim \mathcal{N}(\mathbf{0}, \mathbf{I}_K)}\left[\mathrm{softmax}(\beta(z_k + \mu_k); \beta(\mathbf{z} + \boldsymbol{\mu}))\right],$$

where $\mathrm{softmax}(z_k; \mathbf{z}) = \exp(z_k)/\sum_{l=1}^K \exp(z_l)$. This approximation is accurate for large $\beta$. Second we approximate the expectation using Monte Carlo samples with the reparameterization trick [19]:

$$\mathbb{E}\left[\mathrm{softmax}(\beta(z_k + \mu_k); \beta(\mathbf{z} + \boldsymbol{\mu}))\right] \approx \frac{1}{M}\sum_{i=1}^M \mathrm{softmax}(\beta(\bar{z}_k^i + \mu_k); \beta(\bar{\mathbf{z}}^i + \boldsymbol{\mu})), \tag{5}$$

where $\bar{\mathbf{z}}^i \overset{i.i.d.}{\sim} \mathcal{N}(\mathbf{0}, \mathbf{I}_K)$. With these approximations, we solve for $\boldsymbol{\mu}$ subject to $\mu_1 = 0$ in Eq. (6),

$$\frac{1}{M}\sum_{i=1}^M \mathrm{softmax}(\beta(\bar{z}_k^i + \mu_k); \beta(\bar{\mathbf{z}}^i + \boldsymbol{\mu})) = p_k, \tag{6}$$

for $k = 2, \ldots, K$. The modified Powell method implemented in `SciPy` computes an accurate solution.

**Optimization hyperparameter selection.** Both $M$ and $\beta$ are easy to select as larger values always improve approximation accuracy: larger $M$ leads to better MC approximation and larger $\beta$ leads to better argmax probability approximation. We use $M = 5000$ and $\beta = 1000$ as the default in our experiments, and find this setting works well across all our experiments. Theoretically, a large $M$ is needed when the number of categories $K$ is large for accurate MC approximation; conversely, when

$K$ is small $\beta$ must be large for accurate argmax probability approximation. However, in practice, it is rare to have $K$ larger than 50. Also, for very rare categories, e.g., those with probability $< 10^{-4}$, the limited samples may not suffice to learn correlations; one solution is to drop rare categories or merge rare categories into larger groups. Our default setting achieves satisfying accuracy in synthetic experiment variants even with $K = 50$ and tiny category probabilities (as low as $10^{-4}$).

**Gaussian-max versus Gumbel-softmax.** Both the Gaussian-max and the Gumbel-softmax distributions models a categorical variable taking value in $K$ categories as $x = \mathrm{argmax}(\mathbf{z} + \boldsymbol{\mu})$ for some $\boldsymbol{\mu} \in \mathbb{R}^K$ with independent identically distributed entries of $\mathbf{z}$. The difference lies in the underlying distribution of the latent continuous $\mathbf{z}$: the Gumbel-softmax uses the Gumbel distribution, while we use the Gaussian distribution. Different choices serve for different goals. Using the Gaussian distribution, it is easy to model and estimate the joint distribution for categorical variables as well as their mix with ordered variables. Using the Gumbel distribution, we can write down the closed form expression of $\boldsymbol{\mu}$ in Eq. (2): $\mathbb{P}(\mathrm{argmax}(\mathbf{z} + \boldsymbol{\mu}) = k) = \mathrm{softmax}(\mu_k; \boldsymbol{\mu})$ and $\mu_k = \log p_k$. Thus it is simple to compute the gradients of the argmax probability $\mathbb{P}(\mathrm{argmax}(\mathbf{z} + \boldsymbol{\mu}) = k)$ w.r.t. its parameter, making it widely used in neural networks [16].

### 2.3.2 Copula correlation estimation

Now we show how to find the maximum likelihood estimator (MLE) of copula correlation $\Sigma$, which models the multivariate dependence structure. Given i.i.d. samples $\mathbf{x}^i \overset{\mathrm{iid}}{\sim} \mathrm{EGC}(\Sigma, \mathbf{f}^{\mathrm{ord}}, \boldsymbol{\mu})$ observed at $\mathcal{O}_i$ and missing at $\mathcal{M}_i$ for $i = 1, \ldots, n$. We maximize the observed likelihood by integrating over the latent space that maps to the observation $\mathbf{x}^i_{\mathcal{O}_i}$, as in Eq. (7):

$$\ell_{\mathrm{obs}}(\Sigma; \mathbf{x}^i_{\mathcal{O}_i}) = \frac{1}{n} \sum_{i=1}^{n} \int_{\mathbf{z}^i_{[\mathcal{O}_i]} \in \mathbf{f}^{-1}_{\mathcal{O}_i}(\mathbf{x}^i_{\mathcal{O}_i})} \phi(\mathbf{z}^i_{[\mathcal{O}_i]}; \mathbf{0}, \Sigma_{\mathcal{O}_i, \mathcal{O}_i}) \, d\mathbf{z}^i_{[\mathcal{O}_i]}. \tag{7}$$

The form of this likelihood matches that of the Gaussian copula for ordered variables [35] except that the latent integration is in dimensions $[\mathcal{O}_i]$ instead of $\mathcal{O}_i$. Hence the Expectation Maximization (EM) algorithm in [35] also works in our case by integrating over the appropriate latent indices. Concretely, for the E-step at iteration $t + 1$, we first compute the expectation of the joint likelihood of $(\mathbf{z}^i, \mathbf{x}^i_{\mathcal{O}_i})$ over the distribution of $\mathbf{z}^i$ conditional on the observation $\mathbf{x}^i_{\mathcal{O}_i}$ and the previous correlation estimate $\Sigma^{(t)}$ using the Gaussianity of $\mathbf{z}^i$, denoted as $Q(\Sigma; \Sigma^{(t)})$:

$$Q(\Sigma; \Sigma^{(t)}) = c - \frac{1}{2} \left( \log \det(\Sigma) + \mathrm{tr} \left( \Sigma^{-1} \frac{1}{n} \sum_{i=1}^{n} \mathbb{E}[\mathbf{z}^i(\mathbf{z}^i)^\top | \mathbf{x}^i_{\mathcal{O}_i}, \Sigma^{(t)}] \right) \right), \tag{8}$$

where $c$ is a constant. Then we compute $\mathrm{argmax}_\Sigma Q(\Sigma; \Sigma^{(t)})$ in the M-step, which is the expected "empirical covariance matrix" of $\mathbf{z}^i$. We update $\Sigma^{(t+1)}$ to its corresponding correlation matrix to satisfy the copula parameter constraint similar to [12, 35]:

$$\Sigma^{(t+1)} \leftarrow P_{\mathrm{cor}} \left( \frac{1}{n} \sum_{i=1}^{n} \mathbb{E}[\mathbf{z}^i(\mathbf{z}^i)^\top | \mathbf{x}^i_{\mathcal{O}_i}, \Sigma^{(t)}] \right), \tag{9}$$

where $P_{\mathrm{cor}}(\Sigma)$ returns the correlation matrix corresponding to a covariance matrix $\Sigma$. The EM algorithm is guaranteed to strictly increase the likelihood in Eq. (7) and converges to a stationary point [23, Chapter 3]. The main computation is the expected sample covariance of $\mathbf{z}^i$. Dropping the row index, evaluating $\mathbb{E}[\mathbf{z}\mathbf{z}^\top | \mathbf{x}_{\mathcal{O}}, \Sigma^{(t)}]$ requires computing $\mathbb{E}[\mathbf{z}_{[\mathcal{O}]} | \mathbf{x}_{\mathcal{O}}, \Sigma^{(t)}]$ and $\mathrm{Cov}[\mathbf{z}_{[\mathcal{O}]} | \mathbf{x}_{\mathcal{O}}, \Sigma^{(t)}]$. We give details in the supplement and show how to evaluate the reduced quantities in Section 2.3.3.

Recall the correlation $\Sigma$ must satisfy the constraint in Definition 1: the submatrix $\Sigma_{[j],[j]}$ for each categorical index $j$ must be the identity. The correlation computed by EM generally does not satisfy this constraint. Instead, we use the function $P_{\mathrm{cat}}(\Sigma)$ to correct the correlation $\Sigma$:

$$\Sigma \leftarrow P_{\mathrm{cat}}(\Sigma) = A\Sigma A^\top \text{ where } A = \mathrm{diag}(\Sigma^{-1/2}_{[1],[1]}, \ldots, \Sigma^{-1/2}_{[p_{\mathrm{cat}}],[p_{\mathrm{cat}}]}, \mathbf{I}_{p_{\mathrm{ord}}}), \tag{10}$$

by recognizing that $\mathrm{Cov}[\Sigma^{-1/2}_{[j],[j]} \mathbf{z}_{[j]}] = \mathbf{I}_K$ for each categorical index $j$, when $\mathbf{x}_{\mathrm{cat}}$ has $p_{\mathrm{cat}}$ entries and $\mathbf{x}_{\mathrm{ord}}$ has $p_{\mathrm{ord}}$ entries.

---

**Algorithm 2** Estimation algorithm for the extended Gaussian Copula

---
1: **Input:** Partial observation $\{\mathbf{x}_{\mathcal{O}_i}^i\}_{i=1}^n$, correlation initialization $\Sigma^{(0)}$, $M = 5000, \beta = 1000$.
2: **for** each variable index $j$ **do**                                                                      ▷ Marginal estimation starts
3:      Collect all observed data: $\mathbf{X}_j = \{x_j^i : j \in \mathcal{O}_i, i = 1, \ldots, n\}$
4:      **if** $x_j$ is categorical **then**
5:          Solve Eq. (6) for $\hat{\boldsymbol{\mu}}$ with input $M, \beta$ and $p_k$ equals to the frequency of "$k$" in $\mathbf{X}_j$.
6:          Compute $\hat{f}_j(\cdot; \boldsymbol{\mu}_{[j]})$ in Eq. (3) with $\boldsymbol{\mu}_{[j]}$ as the solved solution $\hat{\boldsymbol{\mu}}$.
7:      **else**
8:          Compute $\hat{f}_j = \hat{F}_j^{-1} \circ \Phi$ where $\hat{F}_j^{-1}$ is the empirical quantile function of $\mathbf{X}_j$.
9:      **end if**
10: **end for**                                                                                              ▷ Marginal estimation ends
11: **for** $t = 0, 1, 2, \ldots$ until convergence **do**                                                    ▷ EM algorithm starts
12:      Compute $\mathbb{E}[\mathbf{z}^i(\mathbf{z}^i)^\top | \mathbf{x}_{\mathcal{O}_i}^i, \Sigma^{(t)}, \hat{\mathbf{f}}]$ for $i = 1, \ldots, n$.
13:      Update $\Sigma^{(t+1)}$ as in Eq. (9).
14:      Correct $\Sigma^{(t+1)} \leftarrow P_{\text{cat}}(\Sigma^{(t+1)})$ as in Eq. (10).
15: **end for**                                                                                              ▷ EM algorithm ends
16: **Output:** marginal estimation $\hat{\mathbf{f}}$, correlation estimation $\hat{\Sigma} = \Sigma^{(t+1)}$.

---

**Computational cost.** Algorithm 2 summarizes the complete estimation algorithm for the EGC model. Marginal estimation only takes a small fraction of total runtime (less than $4\%$ in all experiments of this paper). We observe that the EM algorithm mostly converges in less than 50 iterations. Each iteration has time complexity $O(\alpha n d^3)$ where $\alpha$ denotes the observed entry ratio, $n$ denotes the number of samples and $d = p_{\text{ord}} + p_{\text{cat}} K$ denotes the latent dimension. Line 12 of Algorithm 2 takes the vast majority of the computation time. There are many ways to accelerate the computation. For large $n$, we can parallelize the computation in line 12 of Algorithm 2 or conduct minibatch EM training as in [33]. The minibatch EM performs model updates using randomly sampled data batch and achieves significant speedup without sacrificing accuracy. For large $d$, we can reduce the cubic time complexity in terms of $d$ to linear by imposing a low rank structure on $\Sigma$ following [34], that is $\Sigma = WW^\top + \sigma^2 \mathbf{I}_d$ where $W \in \mathbb{R}^{d \times k}$ with $k \ll d$ and $\sigma^2 \in (0, 1)$. See Appendix B for details.

### 2.3.3 Truncated Gaussian with categorical variables

Finally, we show how to estimate moments and sample from $\mathbf{z}_{[\mathcal{O}]} | \mathbf{x}_{\mathcal{O}}$, as these operations are required for imputation and correlation estimation. Note $\mathbf{z}_{[\mathcal{O}]} | \mathbf{x}_{\mathcal{O}}$ follows a truncated Gaussian distribution truncated into the region $\mathbf{f}_{\mathcal{O}}^{-1}(\mathbf{x}_{\mathcal{O}})$. For each ordered variable with index $j \in \mathcal{O}$, $f_j^{-1}(x_j)$ is an interval. (For continuous $x_j$, this interval is a single point.) Thus without observed categoricals, $\mathbf{f}_{\mathcal{O}}^{-1}(\mathbf{x}_{\mathcal{O}})$ is a Cartesian product of intervals. We call this distribution the *interval-truncated Gaussian*, for which accurate sampling [2] and effective moment estimation [35] methods have been developed.

The truncation region is no longer a Cartesian product of intervals with observed categoricals. Fortunately, for any $\mathbf{x}_{\mathcal{O}}$ we can find an involution $A_{\mathbf{x}}$ (i.e., $A_{\mathbf{x}}^2 = \mathbf{I}$) so that $A_{\mathbf{x}} \mathbf{z}_{[\mathcal{O}]} | \mathbf{x}_{\mathcal{O}}$ follows an interval-truncated Gaussian distribution. Consequently,

$$\mathbb{E}[\mathbf{z}_{[\mathcal{O}]} | \mathbf{x}_{\mathcal{O}}] = A_{\mathbf{x}} \mathbb{E}[A_{\mathbf{x}} \mathbf{z}_{[\mathcal{O}]} | \mathbf{x}_{\mathcal{O}}], \quad \text{Cov}[\mathbf{z}_{[\mathcal{O}]} | \mathbf{x}_{\mathcal{O}}] = A_{\mathbf{x}} \text{Cov}[A_{\mathbf{x}} \mathbf{z}_{[\mathcal{O}]} | \mathbf{x}_{\mathcal{O}}] A_{\mathbf{x}}^\top. \quad (11)$$

Similarly, to sample $\mathbf{z}_i \overset{\text{iid}}{\sim} \mathbf{z}_{[\mathcal{O}]} | \mathbf{x}_{\mathcal{O}}$, we can first sample $\tilde{\mathbf{z}}_i \overset{\text{iid}}{\sim} A_{\mathbf{x}} \mathbf{z}_{[\mathcal{O}]} | \mathbf{x}_{\mathcal{O}}$ and then use $\mathbf{z}_i = A_x \tilde{\mathbf{z}}_i$. To derive $A_{\mathbf{x}}$, first consider that when a categorical variable $x$ takes value $k$, by Eq. (3) we have:

$$f^{-1}(x; \boldsymbol{\mu}) = \{\mathbf{z} \in \mathbb{R}^K : \text{argmax}(\mathbf{z} + \boldsymbol{\mu}) = k\} = \{\mathbf{z} \in \mathbb{R}^K : z_j + \mu_j < z_k + \mu_k, \text{ for } j \neq k\}, \quad (12)$$

Define $\tilde{\mathbf{z}} = A_k \mathbf{z}$ and $\tilde{\boldsymbol{\mu}} = A_k \boldsymbol{\mu}$ where $A_k = -\mathbf{I}_K + \sum_{i=1}^K E_{ik} + E_{kk}$ (matrix $E_{ij}$ has 1 at $(i, j)$-th entry and 0 elsewhere). We can rewrite Eq. (12) to $\{\tilde{\mathbf{z}} : \tilde{z}_j + \tilde{\mu}_j > 0, \text{ for } j \neq k\}$, a Cartesian product of intervals. We can find a matrix $A_j$ as shown above for every categorical variable $x_j$. Let $A_{\mathbf{x}}$ be $\text{diag}(A_1, \ldots, A_{p_{\text{cat}}}, \mathbf{I}_{p_{\text{ord}}})$. Then $A \mathbf{z}_{[\mathcal{O}]}$ follows interval truncated Gaussian and $A_{\mathbf{x}}$ is an involution.

Table 1: Mean(sd) of runtime in seconds and imputation error for each variable type (cat for categorical, cont for continuous, ord for ordinal) over 10 repetitions. See Figure 1 for the error metric.

| $n = 2000$ | Runtime | Cat Error | Cont Error | Ord Error |
|---|---|---|---|---|
| EGC_our | **33(2)** | **0.64(0.01)** | **1.81(0.06)** | **0.50(0.02)** |
| missForest | 53(11) | 0.68(0.01) | 2.06(0.07) | 0.59(0.02) |

| $n = 10000$ | Runtime | Cat Error | Cont Error | Ord Error |
|---|---|---|---|---|
| EGC_our | **107(4)** | **0.64(0.01)** | **1.81(0.04)** | **0.45(0.01)** |
| missForest | 1006(70) | 0.66(0.01) | 2.05(0.04) | 0.52(0.02) |

| $n = 20000$ | Runtime | Cat Error | Cont Error | Ord Error |
|---|---|---|---|---|
| EGC_our | **202(9)** | **0.64(0.01)** | **1.81(0.06)** | **0.42(0.01)** |
| missForest | 3714(267) | 0.66(0.01) | 2.04(0.04) | 0.48(0.01) |

# 3 Experiments

In the experiments, we compare the accuracy of single imputations using our method `EGC` and several competitors. It is more challenging to compare the accuracy for multiple imputation (MI) as MI seeks to recover the correct *distribution* which is mostly unknown in practice. Nevertheless, we designed a synthetic experiment in Appendix C. As shown in Algorithm 2, `EGC` does not require any model hyperparameter but only optimization hyperparamters $M$ and $\beta$, for which we use fixed values $M = 5000$ and $\beta = 1000$ across all our experiments. We show in Section 3.3 that this option already accurately estimates the categorical marginals. All codes are provided in a Github repo[1].

**Competitors.** We implement several imputation methods for mixed categorical and ordered data. Among iterative imputation algorithms, we implement `missForest` [27, 26] and `MICE` [4, 30]. Among LRMC algorithms, we implement `imputeFAMD` [1, 18] and `softImpute` [21, 13]. Both methods one-hot encodes categorical variables. `imputeFAMD` includes special weighting strategy incorporating different variables types, while `softImpute` simply treats the data as numerical. We also implement a `baseline` imputation: majority vote for categorical variables and median imputation for ordered variables. We only need to choose hyperparameters for `imputeFAMD` and `softImpute` to have satisfying performance, for which we further mask 20% of observed training entries as a validation set. Through all experiments in this paper, `softImpute` is the fastest method and `EGC` comes second overall. We include all implementation and runtime details in the supplement.

**Evaluation.** To evaluate an imputation method, we compute misclassification error (ME) for categorical variables and mean absolute error (MAE) for ordered variables at masked entries.

## 3.1 Synthetic experiments

We generate synthetic datasets with 2000 samples and $p \in \{11, 13, 15\}$ variables from the EGC model in Definition 2 with randomly drawn parameters. The $p$ variables include 5 exponentially distributed continuous variables, 5 ordinal variables with five ordinal levels, and $p_{\text{cat}} \in \{1, 3, 5\}$ categorical variables with six categories ($K = 6$). We randomly remove 30% entries of $\mathbf{X}$ for imputation evaluation and generate 10 masked datasets using different seeds.

Fig. 2 shows our method `EGC` performs the best for all variable types in all scenarios. For categorical variables, `EGC` yields a larger improvement with fewer categorical variables. For both continuous and ordinal variables, `EGC` substantially outperforms all other methods. Several methods that impute categorical variables quite well struggle to outperform a simple baseline on ordered variables. The supplement includes more experiments that differ by the number of categories for categorical variables ($K = 3, 9$), the missing ratio (20% and 40%) and missing mechanisms (MAR and MNAR). In general, these experiments show `EGC` performs well for both categorical and ordered variables in mixed data when the model is well-specified. We also ran experiments with much larger sample size and found EGC is much faster compared to missForest, as shown in Table 1.

---

[1] `https://github.com/yuxuanzhao2295/Mixed-categorical-ordered-imputation-extended-Gaussian-copula`

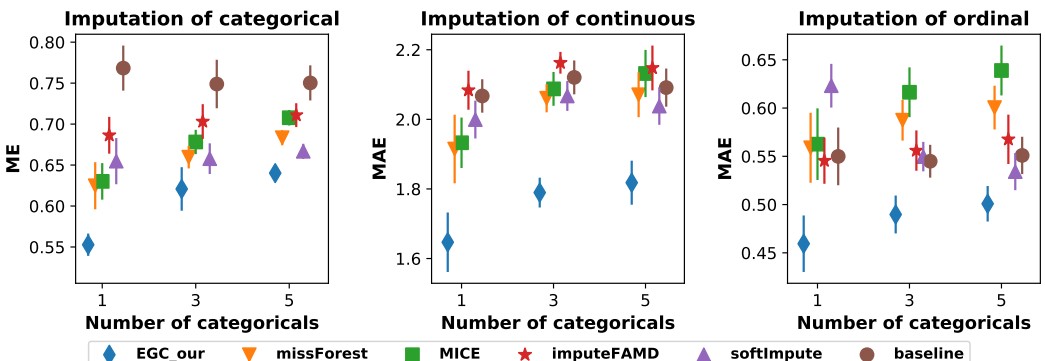

Figure 1: Imputation error on synthetic mixed data with 2000 samples. There are 5 continuous variables, 5 ordinal variables and $1/3/5$ categorical variables with six categories, reported over 10 repetitions (error bars indicate standard deviation).

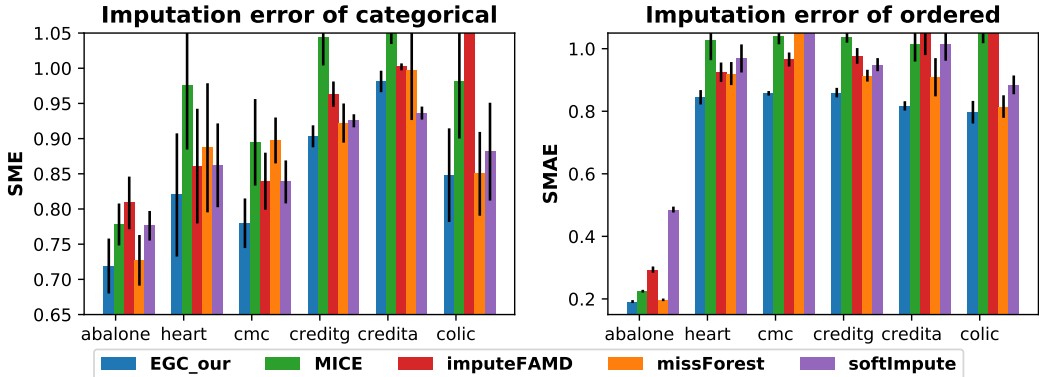

Figure 2: Imputation error of categorical variables and of ordered variables, i.e., ordinal and continuous, on 6 UCI datasets at $20\%$ missingness. Results shown as mean $\pm$ standard deviation.

## 3.2 Real data experiments

We employ six real-world datasets from the UCI machine learning repository here [7]. Each dataset contains both categorical and ordered variables (details in the supplement). We randomly remove $20\%$ of observed entries as a test set to evaluate imputation accuracy and generate 10 masked datasets using different seeds. Since variables in real data often have very different scales, e.g., number of categories for categorical variables and standard deviation for continuous variables, we use an evaluation metric that compensates for the imputation difficulty across variables. For an imputation method, we normalize its imputation error for each variable by the imputation error of `baseline` and then compute the average normalized error for each categorical variable and ordered variable, similar to [35]. We denote the scaled imputation error for categorical variables and ordered variables as SME and SMAE, respectively.

Fig. 2 shows that our method `EGC` is one of the best method in almost all evaluation metrics. `missForest` comes second but its total runtime is four times longer than that of `EGC`. Interestingly, `softImpute` imputes quite accurately for categorical data but poorly for ordered data. The supplement includes more experiments that differ by the missing ratio and missing mechanism, demonstrating that `EGC` consistently outperforms other methods.

## 3.3 Categorical marginal estimation accuracy

Here we verify that `EGC` can learn arbitrary categorical distribution, i.e. the algorithm in Section 2.3.1 finds the root $\mu$ of Eq. (2) accurately. Concretely, we compare two sides of Eq. (2), where the right side is the empirical category probability and the left side is approximated at the estimated $\mu$ using 10000 random samples. Across all used datasets in Section 3.1 and Section 3.2, Fig. 3 shows the

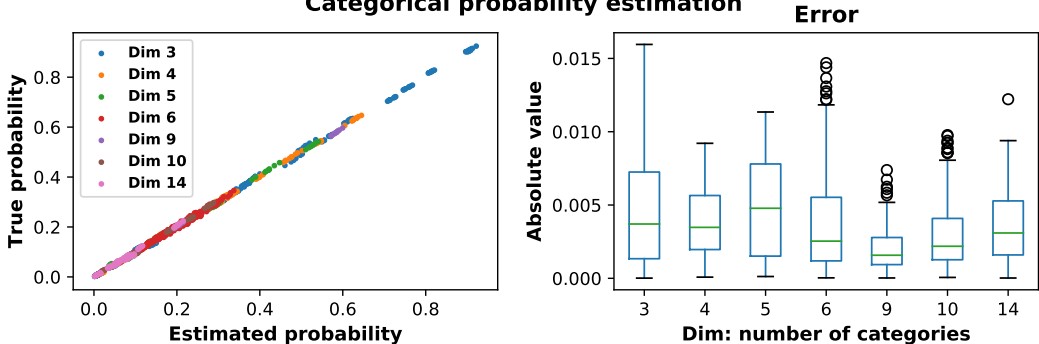

Figure 3: Performance of categorical distribution fitting. The results are collected from 70 datasets: each of 7 datasets (1 synthetic and 6 real data) is randomly masked 10 times. Dim is the number of categories for each categorical variable. *Left* plots the true probability vs the estimated probability of each categorical value, colored by dim, and *Right* plots the absolute values of the estimation error as a boxplot, grouped by dim.

estimated probability indeed matches the true probability well and the performance is robust to the number of categories. Using a larger $M$ in Eq. (6) can further reduce the estimation error, but we find it does not significantly improve the imputation performance.

## 4   Discussion and conclusion

We have presented a probabilistic model, the EGC model, for mixed categorical and ordered data. We developed an imputation method based on the EGC model that empirically demonstrates state-of-the-art performance. Here we discuss two limitations of this paper. First, the estimation algorithm for the EGC model admits guarantees under the MCAR mechanism. Concretely, the marginal estimation requires the MCAR mechanism to accurately match the empirical marginal distribution. Supposing the marginals have been estimated accurately, estimating the correlations consistently is possible under the less restrictive MAR mechanism. While the proposed algorithm in this paper works reasonably well empirically under other missing mechanisms, it may be possible to improve performance by estimating and exploiting the missing mechanism for model estimation. Second, the EGC model uses a Gaussian dependence structure after a marginal transformation. This paper demonstrates the benefits of modeling the marginals explicitly. Learning a more complex dependence structure, e.g., with neural networks, may yield further improvements.

## Acknowledgement

MU and YZ gratefully acknowledge support from NSF Award IIS-1943131, the ONR Young Investigator Program, and the Alfred P. Sloan Foundation. AT is supported by NSF Grants No. DMS-1952757 and No. DMS-2045646 as well as a Simons Fellowship in Mathematics.

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

## A   Online imputation and minibatch training

Here we introduce how to update the parameter of the extended Gaussian copula on a new incomplete data batch, following [33]. For the purpose of online imputation, we update both the marginal and copula correlation at newly observed data. For the purpose of offline minibatch training, we use all available data to estimate the marginal and only update the copula correlation at each data batch.

To update the marginal, we first update the stored observation window, the most recent $m$ (a hyperparameter) observations for each variable, as in [33], then re-compute the empirical category probability, and re-solve the nonlinear equations in Eq. (2) with previous solution as the initialization.

To update the copula correlation $\Sigma^{(t)}$, we first conduct a normal EM iteration on the new data batch and denote the estimation as $\hat{\Sigma}$. Then we perform an incremental update: $\hat{\Sigma}^{(t+1)} = \gamma_t \hat{\Sigma} + (1-\gamma_t)\Sigma^{(t)}$, where $\gamma_t$ is a learning rate in $(0,1)$. The updating rule is a special case on the online EM algorithm in [5]. See [33] for more details.

| Method | Cross Entropy Loss | Runtime |
|--------|--------------------|---------|
| EGC (our) | **0.278 (0.039)** | **14 (1)** |
| MICE | 0.340 (0.004) | 43 (1) |

Table 2: Distribution estimation error of 100 multiple imputation samples on the missing categorical. The error is measured using cross entropy loss (smaller is better), averaged over all missing entries. The runtime is recorded in seconds. Numbers stand for mean (sd) over 10 repetitions.

## B Low rank Gaussian copula for categorical and ordered mixed data

We introduce a special case of the extended Gaussian copula here following the low rank Gaussian copula [34]. Shortly, we assume the latent Gaussian vector $\mathbf{z} \in \mathbb{R}^d$ which generates the data $\mathbf{x}$ follows the probabilistic principal component analysis model [28]:

$$\mathbf{z} = \mathbf{W}\mathbf{t} + \boldsymbol{\epsilon}, \mathbf{t} \sim \mathcal{N}(\mathbf{0}, \mathbf{I}_r), \boldsymbol{\epsilon} \sim \mathcal{N}(\mathbf{0}, \sigma^2 \mathbf{I}_d),$$

where $\mathbf{t}$ and $\boldsymbol{\epsilon}$ are independent and $\mathbf{W} \in \mathbb{R}^{d \times r}$ with $d > r$. Consequently, the copula correlation $\Sigma = \mathbf{W}\mathbf{W}^\top + \sigma^2 \mathbf{I}_d$ contains only $dr + 1$ free parameters. An EM algorithm has been developed in [34] to solve $\mathbf{W}$ and $\sigma^2$. Similar to the results in Section 2.3.2, the EM algorithm for the low rank Gaussian copula also works here by integrating over the appropriate latent indices. The adjustment is to make sure that for a general estimate $\mathbf{W}$ and $\sigma^2$, the formed correlation $\mathbf{W}\mathbf{W}^\top + \sigma^2 \mathbf{I}_d$ satisfies that the submatrix $\Sigma_{[j],[j]} = \mathbf{W}_{[j]}\mathbf{W}_{[j]}^\top + \sigma^2 \mathbf{I}_K$ at each categorical index $j$ must be identity. This requirement implies a constraint in the rank $r$: it must be no smaller than $K$. In other words, the used rank must be no smaller than the largest number of categories among all categorical variables, otherwise $\mathbf{W}_{[j]}\mathbf{W}_{[j]}^\top$ will have rank $r$ and cannot be $(1 - \sigma^2)\mathbf{I}_K$. We can do a correction on $\mathbf{W}$ only ($\sigma^2$ remains unchanged) similar to Eq. (10). That is, at each categorical index $j$, with the singular value decomposition of $\mathbf{W}_{[j]} = \mathbf{U}_j \mathbf{D}_j \mathbf{V}_j^\top$, we correct $\mathbf{W}_{[j]} \leftarrow \mathbf{U}_j \sqrt{1 - \sigma^2} \mathbf{I}_{K,r} \mathbf{V}_j^\top$, where $\mathbf{I}_{k,r} \in \mathbb{R}^{K \times r}$ has 1 in its diagonal.

## C An experiment on multiple imputation

Multiple imputation (MI) is often used in specific case studies ([15] e.g.) where no ground truth is present. There is no generally accepted metric to compare MI methods. Nevertheless, we designed a synthetic experiment scenario to showcase that our MI provides more accurate distribution estimation of the missing entries than MICE using much less time, and MICE is the only other imputation method implemented in this paper that supports MI. For the designed synthetic experiment, the true distribution of missing entries can be accurately approximated. Thus we can compare the estimated distribution provided from MI to the true distribution.

Specifically, we use a synthetic dataset with 5 exponentially distributed continuous variables and 1 categorical variable with 6 categories, 2000 samples and 30% missing ratio, generated similar to that in Section 3.1. For straightforward evaluation, we focus on the categorical variable, for which we use its conditional distribution given all 5 continuous observations as the true distribution. We evaluate the distribution estimation error through the cross entropy loss (smaller is better) between true and estimated category probability, averaged across all missing categorical entries. For both our EGC method and MICE, we take 100 multiple imputation samples and compute the empirical probability for each category as an estimate. The results are reported in Table 2. Our EGC MI achieves significantly smaller loss and smaller runtime than MICE.

