# Supplement: Probabilistic Missing Value Imputation for Mixed Categorical and Ordered Data

**Yuxuan Zhao**
Cornell University
yz2295@cornell.edu

**Alex Townsend**
Cornell University
townsend@cornell.edu

**Madeleine Udell**
Stanford University
udell@stanford.edu

## 1 Methodology supplement

**Notation.** We follow the notations in the main paper. Additionally, we use $E_{ij}$ to denote a matrix which has 1 at the $(i, j)$-th entry and 0 elsewhere.

### 1.1 An identifiable variable of CLG

We first show the copula correlation parameter $\Sigma$ in Definition 1 in the main paper is not identifiable.

**Theorem 1.1.** *For a 2-dimensional categorical vector $(x_1, x_2) \sim \mathrm{CLG}(\Sigma, \boldsymbol{\mu})$, $\mathrm{CLG}(\Sigma^{(kl)}, \boldsymbol{\mu})$ has the same distribution as $\mathrm{CLG}(\Sigma, \boldsymbol{\mu})$ where $\Sigma_{[1],[2]}^{(kl)} = \Sigma_{[1],[2]} + c_1 \sum_{m=1}^{K} E_{km} + c_2 \sum_{m=1}^{K} E_{ml}$ for any constants $c_1, c_2$ and any integers $k, l = 1, \ldots, K$. In other words, adding any constant to a row or column in $\Sigma_{[1],[2]}$ does not change the distribution of $\mathrm{CLG}(\Sigma, \boldsymbol{\mu})$.*

*Proof.* For $(x_1, x_2) \sim \mathrm{CLG}(\Sigma, \boldsymbol{\mu})$, denote the probability that $x_1 = i$ and $x_2 = j$ by $P_{ij}(\Sigma_{[1],[2]}, \boldsymbol{\mu})$ for $i, j = 1, \ldots, K$. To show $\mathrm{CLG}(\Sigma, \boldsymbol{\mu})$ and $\mathrm{CLG}(\Sigma^{(kl)}, \boldsymbol{\mu})$ have the same distribution, it suffices to show that $P_{ij}(\Sigma_{[1],[2]}, \boldsymbol{\mu}) = P_{ij}(\Sigma_{[1],[2]}^{(kl)}, \boldsymbol{\mu})$ for any $i, j = 1, \ldots, K$.

Now define $\Delta_i \in \mathbb{R}^{K-1 \times K}$ such that: (1) the $i$-th column of $\Delta_i$ has all entries equal to $-1$; (2) the remaining $K - 1$ columns of $\Delta_i$ excluding the $i$-th column is $\mathbf{I}_{K-1}$, the identity matrix. Then, we find that

$$i = \mathrm{argmax}(\mathbf{z}_{[1]} + \boldsymbol{\mu}_{[1]}) \iff \Delta_i(\mathbf{z}_{[1]} + \boldsymbol{\mu}_{[1]}) \leq \mathbf{0}.$$

Further, define $\Delta_{i,j} = \mathrm{diag}(\Delta_i, \Delta_j)$, then

$$x_1 = i, x_2 = j \iff \Delta_{i,j}\mathbf{z} + \Delta_{i,j}\boldsymbol{\mu} \leq \mathbf{0}.$$

Note $\Delta_{i,j}\mathbf{z}$ is Gaussian distributed with mean zero and covariance matrix

$$\begin{bmatrix} \Delta_i \Delta_i^\top & \Delta_i \Sigma_{[1],[2]} \Delta_j^\top \\ \Delta_j \Sigma_{[2],[1]} \Delta_i^\top & \Delta_j \Delta_j^\top \end{bmatrix},$$

thus $P_{ij}(\Sigma_{[1],[2]}, \boldsymbol{\mu})$ depends on $\Sigma_{[1],[2]}, \boldsymbol{\mu}$ only through $\Delta_i \Sigma_{[1],[2]} \Delta_j^\top$ and $\Delta_{i,j}\boldsymbol{\mu}$. Now denote $R = \Sigma_{[1],[2]}$ and $R^{(kl)} = R + c_1 \sum_{m=1}^{K} E_{km} + c_2 \sum_{m=1}^{K} E_{ml}$ to simplify the notation. Note the $(s, t)$-th entry of $\Delta_i R \Delta_j^\top$ is $r_{ij} + r_{st} - r_{it} - r_{sj}$ and the $(s, t)$-th entry of $R^{(kl)}$ is $r_{st} + c_1 \mathbb{1}(k = s) + c_2 \mathbb{1}(l = t)$, then it is straightforward that $\Delta_i R \Delta_j^\top = \Delta_i R^{(kl)} \Delta_j^\top$ for arbitrary $c_1, c_2$ and any $k, l = 1, \ldots, K$, which finishes our proof. $\qquad\square$

To eliminate the unidentifiability showed in Theorem 1.1, we provide a variant of CLG with additional constraints in the copula correlation matrix $\Sigma$. Concretely, for each categorical variable $x_j$, we select one dimension in $\mathbf{z}_{[j]}$ to be the base dimension, which does not correlate with any other entry in $\mathbf{z}$. Without loss of generality, we can select the first dimension of $\mathbf{z}_{[j]}$ for all $j$. In other words,

36th Conference on Neural Information Processing Systems (NeurIPS 2022).

$\Sigma_{[j]_1,-[j]_1} = \mathbf{0}$ for any categorical index $j$, where $-[j]_1$ means all indices but $[j]_1$. If $\Sigma$ satisfies this constraint, then $\Sigma^{kl}$ with $\Sigma^{(kl)}_{[1],[2]} = \Sigma_{[1],[2]} + c_1 \sum_{m=1}^{K} E_{km} + c_2 \sum_{m=1}^{K} E_{ml}$ for nonzero $c_1, c_2$ will not satisfy this constraint. With this constraint, there are only $(K-1)^2$ free parameters in $\Sigma_{[j_1],[j_2]}$ for any two categorical variables $x_{j_1}$ and $x_{j_2}$. However, this variant is not permutation-invariant to the labeling of categorical categories.

There are two motivations for this variant. First, as mentioned in Sec 2.1.2 of the main paper, to describe the joint distribution of two categorical variables (each has $K$ categories), $(K-1)^2$ free parameters are sufficient once given the marginal categorical distribution. Second, for a categorical variable $x$ generated as the argmax of a $K$-dim latent Gaussian $\mathbf{z}$, only the difference among the entries of $\mathbf{z}$ matters for the distribution of $x$. In fact, we can even fix the first dimension of $\mathbf{z}$ to be constant 0 and Theorem 1 of the main paper still holds.

## 1.2 Computing the expectation of latent Gaussian

Here we show that computing $\mathbb{E}[\mathbf{z}\mathbf{z}^\top | \mathbf{x}_\mathcal{O}, \Sigma]$ reduces to compute $\mathbb{E}[\mathbf{z}_{[\mathcal{O}]} | \mathbf{x}_\mathcal{O}, \Sigma]$ and $\mathrm{Cov}[\mathbf{z}_{[\mathcal{O}]} | \mathbf{x}_\mathcal{O}, \Sigma]$. By writing $\mathbf{z} = (\mathbf{z}_{[\mathcal{O}]}, \mathbf{z}_{[\mathcal{M}]})$, we first decompose the computation into three parts: (1) $\mathbb{E}[\mathbf{z}_{[\mathcal{O}]}\mathbf{z}_{[\mathcal{O}]}^\top | \mathbf{x}_\mathcal{O}, \Sigma]$; (2) $\mathbb{E}[\mathbf{z}_{[\mathcal{O}]}\mathbf{z}_{[\mathcal{M}]}^\top | \mathbf{x}_\mathcal{O}, \Sigma]$; (3) $\mathbb{E}[\mathbf{z}_{[\mathcal{M}]}\mathbf{z}_{[\mathcal{M}]}^\top | \mathbf{x}_\mathcal{O}, \Sigma]$, and then show each of the three parts can reduce to the mean and covariance of $\mathbf{z}_{[\mathcal{O}]} | \mathbf{x}_\mathcal{O}, \Sigma$. For (1), it is trivial. For (2) and (3), the key technique is the law of total expectation and that

$$\mathbf{z}_{[\mathcal{M}]} | \mathbf{z}_{[\mathcal{O}]} \sim \mathcal{N}\left(\Sigma_{[\mathcal{M}],[\mathcal{O}]}\mathbf{z}_{[\mathcal{O}]}, \Sigma_{[\mathcal{M}],[\mathcal{M}]} - \Sigma_{[\mathcal{M}],[\mathcal{O}]}\Sigma^{-1}_{[\mathcal{O}],[\mathcal{O}]}\Sigma_{[\mathcal{O}],[\mathcal{M}]}\right).$$

Thus

$$\mathbb{E}[\mathbf{z}_{[\mathcal{O}]}\mathbf{z}_{[\mathcal{M}]}^\top | \mathbf{x}_\mathcal{O}, \Sigma] = \mathbb{E}_{\mathbf{z}_{[\mathcal{O}]}}[\mathbf{z}_{[\mathcal{O}]}\mathbb{E}_{\mathbf{z}_{[\mathcal{M}]} | \mathbf{z}_{[\mathcal{O}]}}[\mathbf{z}_{[\mathcal{M}]}^\top | \mathbf{x}_\mathcal{O}, \Sigma]],$$

and

$$\mathbb{E}[\mathbf{z}_{[\mathcal{M}]}\mathbf{z}_{[\mathcal{M}]}^\top | \mathbf{x}_\mathcal{O}, \Sigma] = \mathbb{E}_{\mathbf{z}_{[\mathcal{O}]}}[\mathbb{E}_{\mathbf{z}_{[\mathcal{M}]} | \mathbf{z}_{[\mathcal{O}]}}[\mathbf{z}_{[\mathcal{M}]}\mathbf{z}_{[\mathcal{M}]}^\top | \mathbf{x}_\mathcal{O}, \Sigma]].$$

The remaining computation is straightforward.

## 1.3 Proof of Theorem 1

*Proof.* Denote $\mathbb{P}(\mathrm{argmax}(\mathbf{z} + \boldsymbol{\mu}) = k) = p_k(\boldsymbol{\mu})$ for $k = 1, ..., K$ and $\mathrm{P}(\boldsymbol{\mu}) = (p_1(\boldsymbol{\mu}), ..., p_K(\boldsymbol{\mu}))$. Also define $\mathbf{e}_k$: $\mathbf{e}_k \in \mathbb{R}^K$ has 1 at coordinate $k$ and zero elsewhere.

We prove the existence by contradiction. Suppose there is no satisfying $\boldsymbol{\mu}$. Let

$$\boldsymbol{\mu}^* = \mathrm{argmin}_{\boldsymbol{\mu}} f(\boldsymbol{\mu}), \text{ where } f(\boldsymbol{\mu}) = \sum_{k=1}^{K} |p_k(\boldsymbol{\mu}) - p_k|. \tag{1}$$

Define $I = \{i | p_i(\boldsymbol{\mu}^*) < p_i, i = 1, \ldots, K\}$ and $I^c = \{1, \ldots, K\} - I$. If there is no satisfying $\boldsymbol{\mu}$, then both $I$ and $I^c$ are not empty. Pick an $i \in I$. Since $p_i(\boldsymbol{\mu}^* + \lambda\mathbf{e}_i)$ is a continuous function w.r.t. $\lambda$ and $\lim_{\lambda \to \infty} p_i(\boldsymbol{\mu}^* + \lambda\mathbf{e}_i) = 1$, there exists a $\lambda_0 > 0$ such that $p_i(\boldsymbol{\mu}^* + \lambda_0\mathbf{e}_i) = p_i$. Note $p_k(\boldsymbol{\mu}^* + \lambda\mathbf{e}_i)$ is strictly decreasing w.r.t. $\lambda$ for any $k \neq i$. Thus for $p_k(\boldsymbol{\mu}^* + \lambda_0\mathbf{e}_i) = p_k(\boldsymbol{\mu}^*) - \delta_k$,

we have $\delta_k > 0$ when $k \neq i$ and $\delta_i = p_i(\boldsymbol{\mu}^*) - p_i < 0$. Now

$$
\begin{aligned}
\sum_{k=1}^{K} |p_k(\boldsymbol{\mu}^* + \lambda_0 \mathbf{e}_i) - p_k| &= |p_i(\boldsymbol{\mu}^* + \lambda_0 \mathbf{e}_i) - p_i| + \sum_{k \neq i} |p_k(\boldsymbol{\mu}^* + \lambda_0 \mathbf{e}_i) - p_k| \\
&= \sum_{k \in I - \{i\}} |p_k - p_k(\boldsymbol{\mu}^*) + \delta_k| + \sum_{k \in I^c} |p_k - p_k(\boldsymbol{\mu}^*) + \delta_k| \\
&= \sum_{k \in I - \{i\}} (p_k - p_k(\boldsymbol{\mu}^*) + \delta_k) + \sum_{k \in I^c} |p_k - p_k(\boldsymbol{\mu}^*) + \delta_k| \\
&\leq \sum_{k \in I - \{i\}} (p_k - p_k(\boldsymbol{\mu}^*) + \delta_k) + \sum_{k \in I^c} |p_k - p_k(\boldsymbol{\mu}^*)| + \sum_{k \in I^c} \delta_k \\
&= \sum_{k \in I - \{i\}} |p_k - p_k(\boldsymbol{\mu}^*)| + \sum_{k \in I^c} |p_k - p_k(\boldsymbol{\mu}^*)| + \sum_{k \neq i} \delta_k \\
&= \sum_{k \neq i} |p_k - p_k(\boldsymbol{\mu}^*)| + \sum_{k \neq i} \delta_k \\
&= f(\boldsymbol{\mu}^*) - (p_i - p_i(\boldsymbol{\mu}^*)) + \sum_{k \neq i} \delta_k \\
&= f(\boldsymbol{\mu}^*) + \sum_{k=1}^{K} \delta_k = f(\boldsymbol{\mu}^*)
\end{aligned}
$$

The equality only holds when for each $\in I^c$, $p_k - p_k(\boldsymbol{\mu}^*) \geq 0$, which further leads to $p_k = p_k(\boldsymbol{\mu}^*)$ by the definition of $I^c$. That conflicts with that $I$ is nonempty. Thus we must have

$$
f(\boldsymbol{\mu}^* + \lambda_0 \mathbf{e}_i) < f(\boldsymbol{\mu}^*),
$$

which contradicts our assumption and completes our proof for existence.

Now for uniqueness, assume there exists $\boldsymbol{\mu} \neq \tilde{\boldsymbol{\mu}}$ such that $\mathbf{P}(\boldsymbol{\mu}) = \mathbf{P}(\tilde{\boldsymbol{\mu}})$. Define $I_<, I_=, I_>$ to be the set of entries that $\boldsymbol{\mu}$ is smaller, equal, and larger than $\tilde{\boldsymbol{\mu}}$, respectively. We want to show it must be the case that both $I_<$ and $I_>$ are empty, which contradicts the assumption.

First, if $I_<$ is empty but $I_>$ is not, then for each $i \in I_>$, we define $\boldsymbol{\mu}_i \in \mathbb{R}^K$ such that $\boldsymbol{\mu}_i$ agrees with $\boldsymbol{\mu}$ at all entries but $i$ and agrees with $\tilde{\boldsymbol{\mu}}$ at entry $i$. Since $p_i(\boldsymbol{\mu})$ is strictly increasing w.r.t. $\mu_i$ when fixing $\boldsymbol{\mu}_{\{i\}^c}$, we know $p_i(\boldsymbol{\mu}_i) < p_i(\boldsymbol{\mu})$. Further repeatedly switching one more entry of $\boldsymbol{\mu}_i$ in $I_>$ from $\boldsymbol{\mu}$ to $\tilde{\boldsymbol{\mu}}$ until $I_>$ is exhausted, we have $p_i(\tilde{\boldsymbol{\mu}}_i) < p_i(\boldsymbol{\mu})$, which contradicts the assumption. Similarly we can show the contradiction if $I_<$ is not empty but $I_>$ is empty.

Now consider the case that both $I_<$ and $I_>$ are not empty. Define $\boldsymbol{\mu}^*$ such that $\boldsymbol{\mu}^*$ agrees with $\boldsymbol{\mu}$ over $I_<$ and agrees with $\tilde{\boldsymbol{\mu}}$ over $I_>$. According to above, we know for each $i \in I_>$, $p_i(\boldsymbol{\mu}) > p_i(\boldsymbol{\mu}^*) > p_i(\tilde{\boldsymbol{\mu}})$, which contradicts the assumption. Thus we complete our proof. $\quad\square$

## 2 Experiments supplement

### 2.1 Implementation details

All our implementation codes are provided in a Github repo[1]. All experiments use a laptop with a 3.1 GHz Intel Core i5 processor and 8 GB RAM. All algorithms are implemented using *one core*, although some of them including our `EGC` support parallelism.

The algorithm of `EGC` consists of two parts: the marginal estimation and the correlation estimation. The marginal estimation requires a subroutine to iteratively solve nonlinear systems. We find the available iterative root finding algorithms package in R such as the `rootSolve` do not achieve desired precision occasionally, while the `scipy.optimize.root(method='hybr')` function in Python finds accurate solution in all of our experiments. Through our experiments, `EGC` uses a Python implementation to estimate the marginal and a R implementation to estimate the copula correlation.

---

[1] https://github.com/yuxuanzhao2295/Mixed-categorical-ordered-imputation-extended-Gaussian-copula

Table 1: Algorithm runtime in seconds: mean (sd) over 10 repetitions. The synthetic dataset is under the setting $K = 6, p_{\text{cat}} = 5$.

|           | EGC        | missForest    | MICE       | ImputeFAMD  | softImpute |
|-----------|------------|---------------|------------|-------------|------------|
| Synthetic | 22.0 (1.0) | 58.7 (9.9)    | 82.3 (1.0) | 56.0 (22.3) | 1.4 (0.2)  |
| Abalone   | 9.8 (0.1)  | 136.2 (37.2)  | 2.9 (0.1)  | 46.8 (5.4)  | 0.9 (0.1)  |
| Heart     | 2.1(0.8)   | 2.6 (0.8)     | 4.3 (0.3)  | 11.7 (3.7)  | 0.1 (0.0)  |
| CMC       | 5.5 (1.1)  | 9.2 (1.4)     | 17.3 (0.9) | 41.2 (16.2) | 0.3 (0.0)  |
| Creditg   | 16.0 (0.9) | 43.5 (12.3)   | 74.4 (1.2) | 38.8 (11.3) | 1.0 (0.1)  |
| Credita   | 10.4 (1.1) | 15.6 (3.2)    | 30.0 (8.6) | 23.8 (12.1) | 0.6 (0.1)  |
| Colic     | 3.5 (0.2)  | 6.9 (2.0)     | 40.0 (15.3)| 18.2 (15.8) | 0.4 (0.1)  |

A complete `R` implementation is available though, and a complete `Python` implementation will be available soon. All other algorithms are completely implemented in `R`.

`MICE` is a multiple imputation method. To derive a single imputation from `MICE`, we pool 5 imputed datasets (majority vote for categorical and mean for ordered). We choose the rank for `imputeFAMD` in the grid of $\{1, 3, 5, 7, 9\}$. We choose the regularization parameter for `softImpute` in an exponentially decaying path of length 10 from $0.1 \times \lambda_0$ and $0.99 \times \lambda_0$, where $\lambda_0$ is computed using the provided function `lambda0()` in the package `softImpute`.

```
exp(seq(from=log(lam0*0.99),to=log(lam0*0.1),length=10))
```

The runtime comparison among algorithms are reported in Table 1. For `imputeFAMD` and `softImpute`, the reported time is the total runtime under all searched hyperparameters.

## 2.2 MAR and MNAR mechanism

We conduct additional experiments under a MAR mechanism and a MNAR mechanism to test the sensitivity of implemented imputation methods.

For MNAR, we use a self-masking mechanism which assigns samples different missing probability by their own values for each variable. Concretely, suppose we want to mask $\alpha$ percentage entries as missing, then for each variable, we assign a high missing probability ($\alpha + 10\%$) for samples below the first third quantile, a medium missing probability ($\alpha$) for samples between the first third and the second third quantile, and a low missing probability ($\alpha - 10\%$) for samples above the second third quantile.

For MAR, we first randomly select $1/3$ of variables as observed. Then for each of the remaining $2/3$ of variables, its samples receive three different missing probability similar to the MNAR mechanism but based a randomly selected observed variable instead of its own values. To have $\alpha$ missing ratio and compensate that only $2/3$ of variables may be masked as missing, we use $\frac{3\alpha}{2}$ as the normal missing probability, $\frac{3\alpha}{2} + 10\%$ as the high missing probability, and $\frac{3\alpha}{2} - 10\%$ as the low missing probability.

## 2.3 Synthetic experiments supplement

Fig 1 of the main paper reports the results for categorical variables with six categories under MCAR of $30\%$ missing ratio. In Fig. 1 and Fig. 2, we report the results for different number of categories: three and nine. In Fig. 3 and Fig. 4, we report the results for different missing ratios under MCAR. In Fig. 5 and Fig. 6, we report the results under different missing mechanism (MAR and MNAR). In general, these experiments show EGC performs well for both categorical and ordered variables in mixed data as reported in Section 3.1 of the main paper.

## 2.4 Real data experiments supplement

All used datasets are accessed from `openml.org` through the `R` package `OpenML`. Table 1 provides an overview of the used datasets. The prepared dataset does not distinguish categorical and ordinal variables. We do distinguish them according to the variable description whenever available. Some

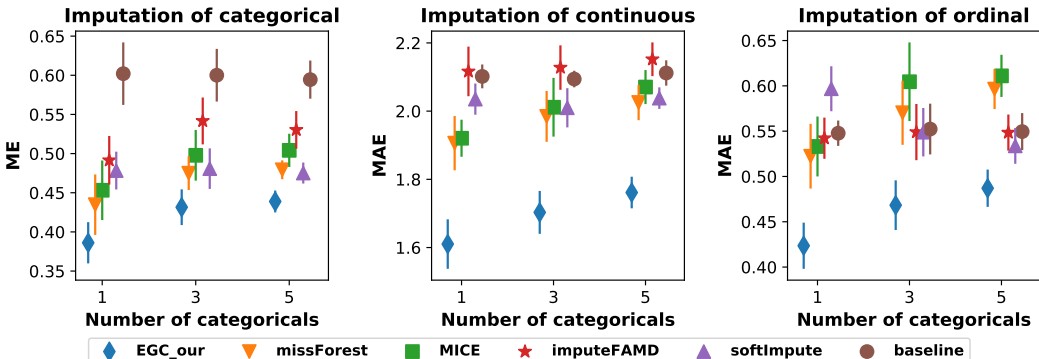

Figure 1: Imputation error on synthetic mixed data under **MCAR of** 30% **missing**. There are 5 continuous variables, 5 ordinal variables and 1/3/5 categorical variables with **three** categories, reported over 10 repetitions (error bars indicate standard deviation).

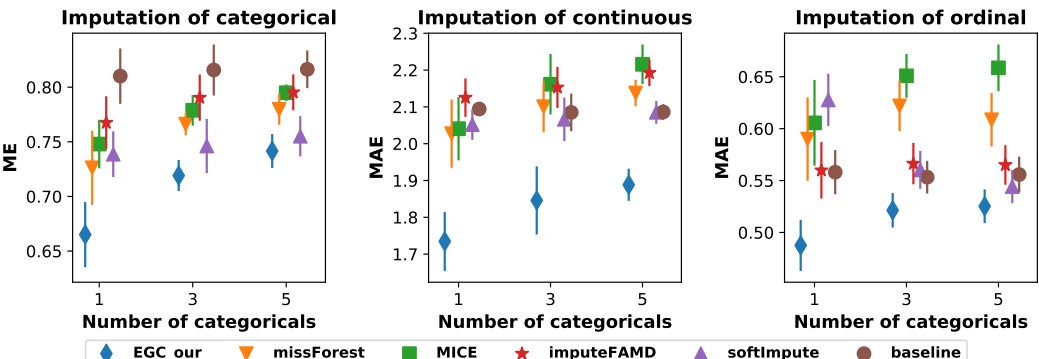

Figure 2: Imputation error on synthetic mixed data under **MCAR of** 30% **missing**. There are 5 continuous variables, 5 ordinal variables and 1/3/5 categorical variables with **nine** categories, reported over 10 repetitions (error bars indicate standard deviation).

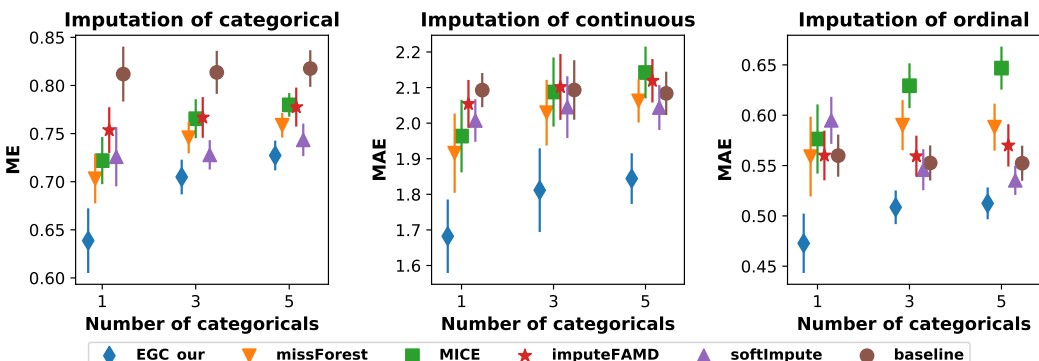

Figure 3: Imputation error on synthetic mixed data under **MCAR of** 20% **missing**. There are 5 continuous variables, 5 ordinal variables and 1/3/5 categorical variables with **six** categories, reported over 10 repetitions (error bars indicate standard deviation).

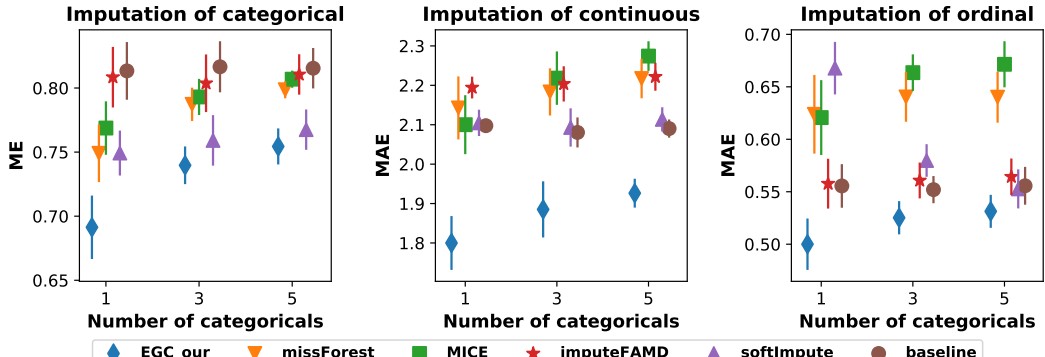

Figure 4: Imputation error on synthetic mixed data under **MCAR of** 40% **missing**. There are 5 continuous variables, 5 ordinal variables and 1/3/5 categorical variables with **six** categories, reported over 10 repetitions (error bars indicate standard deviation).

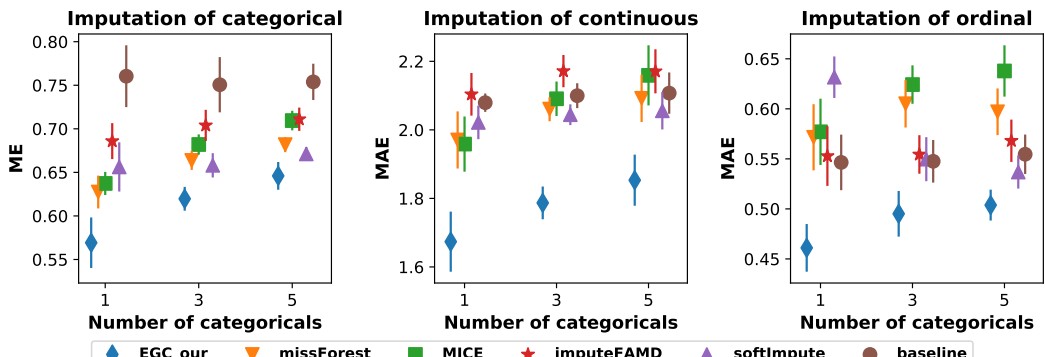

Figure 5: Imputation error on synthetic mixed data under **MAR of** 30% **missing**. There are 5 continuous variables, 5 ordinal variables and 1/3/5 categorical variables with **six** categories, reported over 10 repetitions (error bars indicate standard deviation).

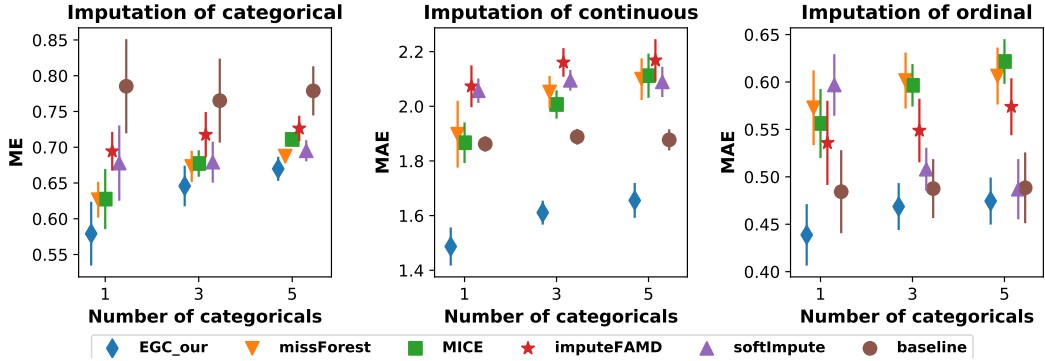

Figure 6: Imputation error on synthetic mixed data under **MNAR of** 30% **missing**. There are 5 continuous variables, 5 ordinal variables and 1/3/5 categorical variables with **six** categories, reported over 10 repetitions (error bars indicate standard deviation).

Table 2: Used UCI dataset overview.

|         | OpenML ID | $n$  | $p_{cat}$ | $p_{ord}$ |
|---------|-----------|------|-----------|-----------|
| Abalone | 1557      | 4177 | 1         | 7         |
| Heart   | 53        | 270  | 3         | 10        |
| CMC     | 23        | 1473 | 1         | 8         |
| Creditg | 31        | 1000 | 8         | 12        |
| Credita | 29        | 690  | 4         | 10        |
| Colic   | 25        | 368  | 4         | 19        |

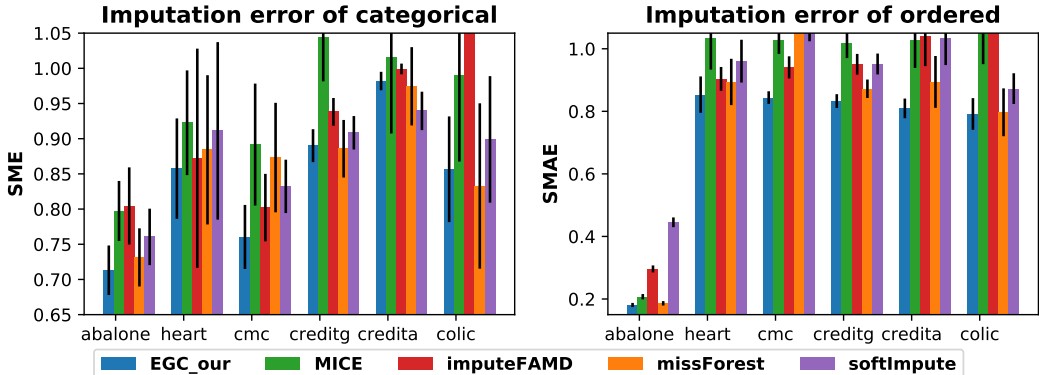

Figure 7: Imputation error of categorical variables and of ordered variables, i.e., ordinal and continuous, on 6 UCI datasets **under MCAR of** $10\%$ **missingness**. Results shown as mean $\pm$ standard deviation.

features are removed because their distribution are highly concentrated at a single value (more than $.95\%$). All preprocessing information are provided in the codes.

Fig 1 of the main paper reports the results under MCAR of $20\%$ missing ratio. In Fig. 7 and Fig. 8, we report the results for different missing ratios under MCAR. In Fig. 9 and Fig. 10, we report the results under different missing mechanism (MAR and MNAR). In general, these experiments show EGC performs well for both categorical and ordered variables in mixed data as reported in Section 3.2 of the main paper.

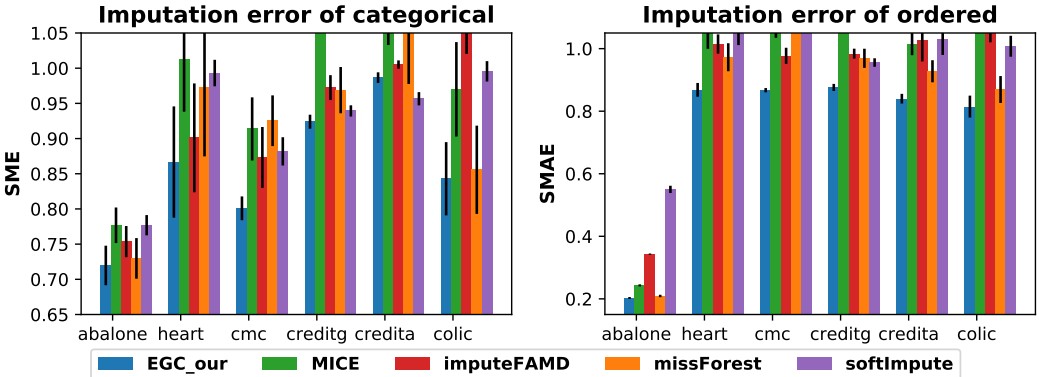

Figure 8: Imputation error of categorical variables and of ordered variables, i.e., ordinal and continuous, on 6 UCI datasets **under MCAR** $30\%$ **missingness**. Results shown as mean $\pm$ standard deviation.

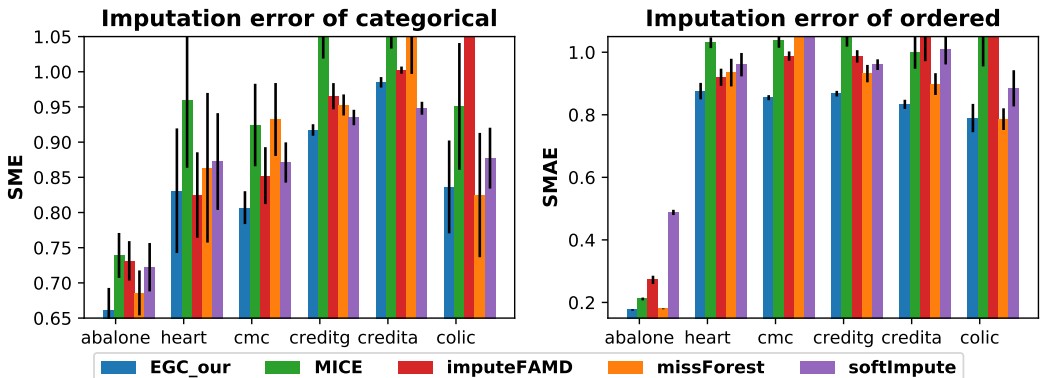

Figure 9: Imputation error of categorical variables and of ordered variables, i.e., ordinal and continuous, on 6 UCI datasets **under MAR** 20% **missingness**. Results shown as mean $\pm$ standard deviation.

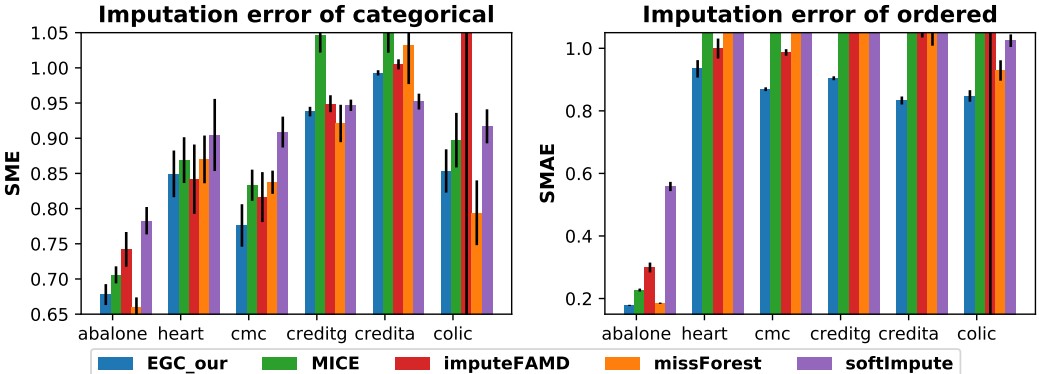

Figure 10: Imputation error of categorical variables and of ordered variables, i.e., ordinal and continuous, on 6 UCI datasets **under MNAR** 20% **missingness**. Results shown as mean $\pm$ standard deviation.