# OpenReview forum: "Probabilistic Missing Value Imputation for Mixed Categorical and Ordered Data"
_NeurIPS.cc/2022/Conference — NeurIPS 2022 Accept_

### Official Review · Reviewer_5kUW · 2022-06-20

**Rating:** 5
**Confidence:** 4
**Soundness:** 3 good
**Presentation:** 3 good
**Contribution:** 2 fair

**Summary:**

This manuscript proposes an extension of the Gaussian copula model for imputing missing values on mixed non-ordinal (e.g., categorical) and ordered (e.g., continuous) data. The authors show the superiority of the proposed method on 1 synthetic and 6 real datasets in comparison to several competing approaches.

**Questions:**

1) in lines 93-94, why the assumption of $\mu_1=0$ is necessary for the model to work?
2) in lines 102-103, why the Gaussian-Max distribution assumption is necessary for $x_j$, and how this assumption affects the generality of the proposed model? Does it always hold? What happens if it does not hold? How can we check in practice whether this assumption holds?
3) how does the proposed method perform when the input data has a very large rank?


**Limitations:**

The authors must include a section to discuss the limitations of the proposed method.

**Strengths And Weaknesses:**

# Strengths
1) The paper addresses an important and well-defined problem in the machine learning context.
2) The text is clear and the technical contributions are well presented.
3) The proposed method shows a substantial improvement over the competing methods.

# Weaknesses
1) The main contribution of this manuscript overlaps substantially with a previously published work "Missing Value Imputation for Mixed Data via Gaussian Copula" (https://dl.acm.org/doi/abs/10.1145/3394486.3403106, which is not even mentioned in the related works), which affects negatively the significance of this work. The authors must i) discuss the added value of their new method and ii) compare its performance with the existing one.
2) Even though some results for the MAR and MNAR scenarios are presented in the supplementary material since all the contributions in the manuscript are made based on the MCAR assumption, the author must discuss thoroughly how violating the MCAR assumption may affect the assumptions of the proposed model.
3) The computational complexity of the method is cubic with respect to the dimension of the latent which is restricting in practice. The authors propose to use a low-rank approximation of $\Sigma$, but the effect of such an approximation is not evaluated in the experimental results.
4) While promised as a method for multiple imputation (lines 63-65), the method is not evaluated for multiple imputation in the experiments (stating that the correct distribution is mostly unknown). Maybe this could be evaluated on synthetic data at least.
5)  While the proposed method has two hyperparameters (referred to as optimization hyperparameters $M$ and $\beta$), the authors claim that their method has no hyperparameters. Further, it is not very clear how the values for these hyperparameters are specified.

---

> ### Author Response · Authors · 2022-08-02
> **Authors' response to review**
>
> Thanks for your detailed comments, which we address below. For your comments listed in the Weakness:
> 1. The previous work [1] is the motivation and building block of our paper. However, **[1] can only be used on ordered data, and cannot be used for any categorical with more than 2 categories**. Their software cannot be used to impute categorical data even when one-hot encoding is applied. That is because their imputation cannot enforce the constraint that among a set of 0/1 binary variables generated by one-hot encoding a categorical variable, only one can be 1. That is why  [1] was not implemented in this paper. There is no obvious way to solve our problem using their provided codes. **Our paper solves the problem of categorical imputation**: using the proposed Gaussian-max distribution for categorical variables, we extend [1] so that it can be applied to datasets with an arbitrary number of categorical variables. Additionally, we take it seriously to acknowledge existing work and we referenced [1] 11 times throughout this paper (2 in introduction and 9 in methodology), to distinguish our original contribution from existing work. We will add the above discussion into the main paper.
>
> 2. The imputation has two parts: first, it estimates a model from partial observations and then imputes the missing entries based on the estimated model. The second step does not make any assumption on the missing mechanism. For our proposed EGC model, the first step has two parts: marginal estimation and correlation estimation. The marginal estimation accurately matches the empirical marginal distribution, so it requires the missing uniformly at random (MCAR) mechanism to be accurate. With an accurately estimated marginal, the correlation estimation only requires missing at random (MAR) mechanism to be consistent, according to [2, Chapter 6.2]. Quantifying the influence of a violated missing mechanism on the resulting imputation is challenging, for almost all imputation methods. We will add this discussion into our main paper.
>
> 3. The cubic time complexity is not a problem for skinny datasets with a few hundred features, as shown in [1, Section 7.3], and skinny datasets are common in practice. Thus we believe our paper makes a useful contribution to the current literature. Scaling to wider datasets is an interesting and important challenge, but it is beyond the scope of this paper. Nevertheless, we described one potential way for our proposed EGC to scale to wider datasets in the supplement (Sec 1.5).
>
> 4. Multiple imputation (MI) is often used in specific case studies ([3] e.g.) where no ground truth is present. There is no generally accepted metric to compare MI methods. Nevertheless, we designed a synthetic experiment scenario, where the true distribution of missing entries can be accurately approximated, to showcase that **our MI provides more accurate distribution estimation of the missing entries than MICE using much less time**, and MICE is the only other imputation method implemented in this paper that supports MI. The experiment and result is put in the appendix A of the revised paper.
>
> 5. We do not treat M and beta as hyperparameters because (1) larger values of M and beta always bring more accurate approximations (Eq 5) and thus also more accurate parameter estimation; (2) we use fixed values of M and beta throughout all experiments in this paper. The values we use are large enough: further increases do not yield significant performance improvements, but merely increases computation time.
>
> For your Q1 and Q2: mu1=0 is a sufficient but not necessary condition for the Gaussian-max distribution to model an arbitrary 1-dimensional categorical distribution. This is the result of Theorem 1, which requires no restrictive assumptions. Assumptions are needed only to ensure the model can match a *multivariate* distribution, as in Definition 1 and 2. In other words, the assumption is only on the multivariate dependence, but not on the marginal distribution of an individual categorical variable. That is, **the EGC model fits arbitrary (ordered and categorical) marginal distributions exactly.** We think that’s awesome! As to whether / when the multivariate assumptions hold: that’s a super interesting question, and definitely worth studying, but beyond the scope of the present paper.
>
> For your Q3:
> Our method does not depend explicitly on the rank of the dataset, unlike methods like softimpute and imputeFAMD. As a result, we expect that the method will perform well (at least relative to these others) on data with high rank.
>
> [1] Zhao, Yuxuan, and Madeleine Udell. "Missing value imputation for mixed data via gaussian copula." KDD 2020.
>
> [2] Roderick JA Little and Donald B Rubin. 2002. Statistical analysis with missing data.
>
> [3] Hollenbach, Florian M., et al. "Multiple imputation using Gaussian copulas." Sociological Methods & Research 2021

---

> > ### Comment · Reviewer_5kUW · 2022-08-10
> > **Thank you for the clarification**
> >
> > I would like to thank the authors for clarification and for performing an extra experiment. I rather increase my score for this paper. I still believe more experimental and theoretical validations for challenging MNAR scenarios are needed to make this work stronger.  Also, I have still some concerns about how $M$ and $\beta$ are selected. The authors mentioned that they took the largest possible value but how this largest value is decided? (on training data on the test data). How the user of your method should decide these values on different datasets?

---

> > > ### Author Response · Authors · 2022-08-10
> > > **Clarification ho selected optimization parameter**
> > >
> > > Thanks for raising your score! We actually selected the value of M and beta on randomly generated category probabilities before we conducted any experiment reported in this paper.  Thus the marginal estimation performance on our selected M and beta are already test data performance!
> > >
> > > You could also refer to our response to reviewer JaBb for more discussion on selected optimization parameter, which I pasted below for your convenience:
> > >
> > > In short, larger M leads to better quality of the Monte Carlo (MC) approximation in Eq (5) and larger beta leads to better argmax probability approximation. Theoretically speaking, very large K requires very large M for accurate MC approximation, and very small category probability requires very large beta for accurate argmax probability approximation. However, in practice, it is rare to have K larger than 50. Also, if there exists a very small category probability (<1e-4), i.e. a very rare category, the limited samples may not be sufficient to learn relationship regarding the rare category. We found our algorithm with default values still has satisfying accuracy in our synthetic experiment variants with K=50 and tiny category probability (as low as 1e-4). Thus our provided default values of M and beta should suffice for most realistic cases.
> > >
> > > If a categorical variable with more than 50 categories or very small category probability (<1e-4) does exist, one may want to preprocess it by merging similar categories and dropping rare categories, but one could also increase M and beta accordingly. The fact that larger M and beta are always better makes its tuning very easy if ever needed.

---

### Official Review · Reviewer_ZBKz · 2022-07-11

**Rating:** 5
**Confidence:** 2
**Soundness:** 2 fair
**Presentation:** 2 fair
**Contribution:** 1 poor

**Summary:**

The authors through this manuscript propose a probabilistic model to impute missing values of mixed data including continuous, categorical and ordinal variables that supports both single and multiple imputation. The proposed model is based on extended Gaussian copula, is free of hyperparameters, and makes no assumptions on the marginal distribution of the data types. Authors run a series of experiments to compare the accuracy of the imputations of the proposed model with several competitors.

**Questions:**

1) The dataset used in Synthetic experiment has 2000 samples, and in Real data experiments a maximum of just 4177 samples. Did the authors test the convergence of the EGC model on a much larger dataset? If yes, what were the drawbacks you noticed with the performance?
2) Also, noticed a minor typographical error at Line 49, "approach can explicitly modeling the categorical distribution"

**Limitations:**

Other than the scalability of the proposed EGC model, authors did address the limitations of the study.

**Strengths And Weaknesses:**

Authors did a good job in articulating the importance of their study and it's contributions to the literature including the shortcomings of the existing imputation methods and the proposed algorithm methodology. The proposed methodology is evaluated through a series of experiments to compare the accuracy with other competing methods. I agree with the authors that several existing methods such as MICE, MissForest etc. are good but converge slowly especially for large datasets, however, I strongly believe as part of their experiments authors should have evaluated real world applicability by testing the performance of the proposed EGC model on a large dataset and test it's scalability.

---

> ### Author Response · Authors · 2022-08-02
> **Authors' response to review**
>
> Thank you for your positive feedback and pointing out the typos, which we will correct.  For your question regarding the scalability, we want to first clarify that each iteration of the model fitting algorithm **scales linearly in terms of the number of samples (n)**, as discussed in lines 220-230. Thus, a larger n is not a problem for using our proposed EGC model. We also added a large data experiment to address your concern empirically.
>
> The added experiment is on the synthetic dataset with 15 variables (5 categorical) in Sec 3.1, but with larger sample sizes 10000 and 20000 (originally 2000 in Sec 3.1). We implement our EGC and missForest here only, since MICE is too slow and low rank methods naturally scale well on large datasets. Table 1 reports the results. The runtime of our EGC increases much slower than missForest and it achieves better imputation performance. **The results here indicate that for large n datasets, missForest can be prohibitively expensive while our EGC still scales well.**
>
> Table 1: Added synthetic experiment: Mean (sd) of runtime in seconds and imputation error for each variable type (cat for categorical, cont for continuous and ord for ordinal), over 10 repetitions. See Figure 1 in the main paper for the error metric.
>
>
> | n = 2000 | Runtime | Cat Error|Cont Error|Ord Error|
> | ------------- |-------------|-------------|-------------|-------------|
> | EGC (our)| **33 (2)** |**0.64 (0.01)**|**1.81 (0.06)**|**0.50 (0.02)**|
> | missForest     | 53 (11) |0.68 (0.01)|2.06 (0.07)|0.59 (0.02)|
> | **n = 10000** |
> | EGC (our)| **107 (4)** |**0.64 (0.01)**|**1.81 (0.04)**|**0.45 (0.01)**|
> | missForest     | 1006 (70) |0.66 (0.01)|2.05 (0.04)|0.52 (0.02)|
> | **n = 20000** |
> | EGC (our)| **202 (9)** |**0.64 (0.01)**|**1.81 (0.03)**|**0.42 (0.01)**|
> | missForest     | 3714 (267) |0.66 (0.01)|2.04 (0.04)|0.48 (0.01)|

---

### Official Review · Reviewer_JaBb · 2022-07-11

**Rating:** 7
**Confidence:** 3
**Soundness:** 3 good
**Presentation:** 3 good
**Contribution:** 3 good

**Summary:**

The authors propose a single and multiple missing value imputation method for mixed data under the MCAR (missing completely at random) assumption. The method is based on using a latent Gaussian distribution in the form of an ordinary Gaussian copula model for ordered data (ordinal and continuous) and an *extended* Gaussian copula model proposed by the authors for categorical variables. The author showcase impressive results for their approach, which outperforms standard imputation methods in both synthetic and real-world experiments.

**Questions:**

Would it make sense to relax the constraint for the covariance of **z** in (1), for instance if there are variables that are partially ordered?

**Ethics Review Area:**

["I don’t know"]

**Limitations:**

There are no potential negative societal impacts for this work that I can imagine.

**Strengths And Weaknesses:**

This work represents a novel and sound method for handling single and multiple imputation for mixed data. The extended Gaussian copula probabilistic model using the argmax transformation is, to the best of my knowledge, novel. The results showcased are impressive and suggest that the work may have significant impact. Finally, the paper is almost flawlessly written and clearly structured.

Minor comments:
- p1 (page 1), l4 (line 4): Insert 'this is' before "challenging".
- p1, l19: "we" needs to be capitalized.
- p5, l189: "expection" $\to$ 'expectation'

---

> ### Author Response · Authors · 2022-08-02
> **Authors' response to review**
>
> Thank you for your positive feedback and for pointing out the typos, which we will correct. For your question, we want first to clarify that it is okay to use a less restricted covariance of z in (1), see [1]. However, that will introduce redundant parameters. One major contribution we make is the observation that an identity covariance for (1) suffices to model any univariate categorical distribution (Thm 1). Relaxing to partially ordered z is a fascinating point, and it may be used to generate variables of special types. We consider it outside the scope of this paper as partially ordered z is not naturally suited for modeling a categorical variable.
>
> [1] Christoffersen, Benjamin, et al. "Asymptotically Exact and Fast Gaussian Copula Models for Imputation of Mixed Data Types." Asian Conference on Machine Learning. PMLR, 2021.

---

> > ### Comment · Reviewer_JaBb · 2022-08-09
> > **Thanks**
> >
> > Thank you for clarifying the point regarding the covariance of **z** in Equation (1).
> >
> > I think reviewer **5kUW** makes a good point regarding the claim that no hyperparameter tuning is necessary. Are you claiming that the optimization hyperparameters never need to be changed, i.e., $M = 5000, \beta=1000$ will suffice whatever the size of the problem (K) is?
> >
> > I found one more typo in line 251: "hyperparamters".
> >
> > Apart from that, I remain positive about the paper, having read the other reviews and the rebuttal.

---

> > > ### Author Response · Authors · 2022-08-10
> > > **Regarding optimization hyperparameter**
> > >
> > > Thanks for your question. We add more discussion here to address the concern regarding optimization hyperparameter and will add it to our revised paper.
> > >
> > > In short, larger M leads to better quality of the Monte Carlo (MC) approximation in Eq (5) and larger beta leads to better argmax probability approximation. Theoretically speaking, very large K requires very large M for accurate MC approximation, and very small category probability requires very large beta for accurate argmax probability approximation. However, in practice, it is rare to have K larger than 50. Also, if there exists a very small category probability (<1e-4), i.e. a very rare category, the limited samples may not be sufficient to learn relationship regarding the rare category. **We found our algorithm with default values still has satisfying accuracy in our synthetic experiment variants with K=50 and tiny category probability (as low as 1e-4). Thus our provided default values of  M and beta should suffice for most realistic cases.**
> > >
> > > If a categorical variable with more than 50 categories or very small category probability (<1e-4) does exist, one may want to preprocess it by merging similar categories and dropping rare categories, but one could also increase M and beta accordingly. The fact that larger M and beta are always better makes its tuning very easy if ever needed.

---

### Meta-Review · Area_Chair_fm5Z · 2022-08-27

**Recommendation:** Accept
**Confidence:** Less certain

**Metareview:**

The authors propose a single and multiple missing value imputation method for mixed data under the MCAR (missing completely at random) assumption. The method is based on using a latent Gaussian distribution in the form of an ordinary Gaussian copula model for ordered data (ordinal and continuous) and an extended Gaussian copula model proposed by the authors for categorical variables. The author showcase impressive results for their approach, which outperforms standard imputation methods in both synthetic and real-world experiments.

The reviewers agree that this is overall solid work that makes important technical advances and illustrates the usefulness of the approach. The authors were able to address the main concerns by the reviewers in the discussion. There are some remaining questions, but they appear to be not of a nature that would generally question the results or the overall contribution. Moreover, reviewer JaBb quite strongly supports the acceptance of the manuscript.

Taken together, I think this manuscript is a very good submission and I support it's acceptance.


**Award:**

No

---

### Decision · Program_Chairs · 2022-09-14

Accept